# The Deep Proteomics Approach Identified Extracellular Vesicular Proteins Correlated to Extracellular Matrix in Type One and Two Endometrial Cancer

**DOI:** 10.3390/ijms25094650

**Published:** 2024-04-24

**Authors:** Valeria Capaci, Feras Kharrat, Andrea Conti, Emanuela Salviati, Manuela Giovanna Basilicata, Pietro Campiglia, Nour Balasan, Danilo Licastro, Federica Caponnetto, Antonio Paolo Beltrami, Lorenzo Monasta, Federico Romano, Giovanni Di Lorenzo, Giuseppe Ricci, Blendi Ura

**Affiliations:** 1Institute for Maternal and Child Health—IRCCS Burlo Garofolo, 65/1 Via dell’Istria, 34137 Trieste, Italyferas.kharrat@burlo.trieste.it (F.K.); andrea.conti@burlo.trieste.it (A.C.); nour.balasan@burlo.trieste.it (N.B.); federico.romano@burlo.trieste.it (F.R.); giovanni.dilorenzo@burlo.trieste.it (G.D.L.); giuseppe.ricci@burlo.trieste.it (G.R.); blendi.ura@burlo.trieste.it (B.U.); 2Department of Pharmacy, University of Salerno, 84084 Salerno, Italy; esalviati@unisa.it (E.S.); pcampiglia@unisa.it (P.C.); 3Department of Advanced Medical and Surgical Sciences, University of Campania “Luigi Vanvitelli”, 80138 Naples, Italy; manuelagiovanna.basilicata@unicampania.it; 4AREA Science Park, Basovizza, 34149 Trieste, Italy; licastrod@gmail.com; 5Department of Medicine, University of Udine, 33100 Udine, Italy; federica.caponnetto@uniud.it (F.C.); antonio.beltrami@uniud.it (A.P.B.); 6Azienda Sanitaria Universitaria Friuli Centrale, 33100 Udine, Italy; 7Department of Medicine, Surgery and Health Sciences, University of Trieste, 34149 Trieste, Italy

**Keywords:** proteomics, mass spectrometry, extracellular matrix

## Abstract

Among gynecological cancers, endometrial cancer is the most common in developed countries. Extracellular vesicles (EVs) are cell-derived membrane-surrounded vesicles that contain proteins involved in immune response and apoptosis. A deep proteomic approach can help to identify dysregulated extracellular matrix (ECM) proteins in EVs correlated to key pathways for tumor development. In this study, we used a proteomics approach correlating the two acquisitions—data-dependent acquisition (DDA) and data-independent acquisition (DIA)—on EVs from the conditioned medium of four cell lines identifying 428 ECM proteins. After protein quantification and statistical analysis, we found significant changes in the abundance (*p* < 0.05) of 67 proteins. Our bioinformatic analysis identified 26 pathways associated with the ECM. Western blotting analysis on 13 patients with type 1 and type 2 EC and 13 endometrial samples confirmed an altered abundance of MMP2. Our proteomics analysis identified the dysregulated ECM proteins involved in cancer growth. Our data can open the path to other studies for understanding the interaction among cancer cells and the rearrangement of the ECM.

## 1. Introduction

Among gynecological cancers, endometrial cancer (EC) is the most common in developed countries, with an estimated 417,000 new cases and 97,000 deaths worldwide in 2020 [1]. Most cases occur between 55 and 70 years of age [2]. Several risk factors for EC have been identified, including obesity, states of unopposed estrogen, early menarche, or late menopause, as well as hereditary cancer syndromes such as Lynch and Cowden’s syndrome [3]. EC is classified into two major categories: type I and type II. Type I is the most common endometrial cancer (70% to 80%). It includes low-grade tumors (G1 and G2) usually linked to an excess of estrogen stimulation and is strongly associated with obesity and other elements included in the metabolic syndrome [4]. However, EC patients with type I are usually diagnosed in the early stages and have a favorable prognosis (85% 5-year overall survival). Type II ECs include high-grade non-endometrioid adenocarcinomas (20%), uterine clear cell carcinomas (1 to 5%), and uterine carcinosarcomas (2 to 5%). EC patients with the second pathogenetic type are often diagnosed at advanced stages, present poorly differentiated tumors, a higher propensity to deep invasion of the tumor into the myometrium, a higher frequency of metastatic spread into the pelvic lymph nodes, and doubtful prognosis (the 5-year overall survival rate between 36% and 80%) [4].

In recent years, findings have shown that the tumor microenvironment plays a crucial role in cancer development, progression, and in the clinical outcome [5,6]. Indeed, ECs, as with many other tumor types, behave as complex organ-like systems in which cancer cells undergo dynamic interactions with stomal cells and with the extracellular matrix, shaping the tumor microenvironment (TME). These interactions can both support cancer growth and survival and, vice versa, limit it, triggering an immune response [7]. In particular, ECM components confer unique physical, biochemical, and biomechanical properties that are essential for regulating cell behavior [8]. Interestingly, cancer cells secrete a plethora of proteins (i.e., cytokines, chemokines, growth factors, proteases, and ECM components), both free or loaded in extracellular vesicles (EVs), able to remodel the ECM and determine a supportive pro-tumorigenic microenvironment.

Extracellular vesicles (EVs) are cell-derived membrane-surrounded vesicles and contain different bioactive molecules. They exert several roles in the physiological processes, varying from cellular communication to immune response and apoptosis [9]. Beyond their diverse physiological roles, EVs can be important biomarkers for several pathological conditions, including different types of cancer [10]. Several studies highlighted the role of EVs in endometrial cancer [11,12,13] and other female reproductive dysfunctions [13]. EVs might contain different bioactive materials like proteins, metabolites, lipids, and nucleic acids.

Interestingly, EVs can directly interact with the ECM, actively contributing to its remodeling, degrading the matrix with their surface-associated enzymes, or promoting the release of ECM-degrading proteins, finally promoting tumor cell invasion and metastasis. [14]. Among EVs, the best studied and characterized are the exosomes. Exosomes are small vesicles (30–120 nm) originating in cells from multivesicular bodies (MVBs) and secreted in many body fluids. They are able to transport a large variety of nucleic acids and proteins, influencing the microenvironment and receiving cells at distal sites in the body [15].

The use of high-resolution mass spectrometry (HRMS) in proteomics is helping to understand the molecular mechanisms of tumors [16]. In addition, MS has been successfully used in the identification of biomarkers in biological fluids in the EC [17]. Interestingly, by targeted proteomics, Mariscal et al. found an increase in the adhesion protein LGALS3BP levels from a cohort of EC patients with a high risk of recurrence, highlighting its potential in liquid biopsy [11].

The LC-MS/MS (liquid chromatography with tandem mass spectrometry) based proteomics mainly uses two approaches in data acquisition—the DDA (data dependent acquisition) and DIA (data-independent acquisition). Both acquisitions are making a great contribution to understanding the pathogenesis of tumors and identifying new biomarkers [18].

In this study, by using a deep proteomics approach in the EVs of different cellular models, we aimed to characterize the EV proteome in order to identify dysregulated ECMs and associated proteins.

## 2. Results

### 2.1. Extracellular Vesicle Characterization

In order to study EVs from EC, we exploited four different EC cell lines, namely EC type I (Ishikawa, AN3CA) and type 2 (HEC1A, KLE), respectively, of primary and metastatic origin. From the conditioned cell line medium, we isolated EVs and performed a biophysical and biological characterization. First, we measured the EVs concentration and dimension by NTA analysis. NTA measurements of AN3CA (Figure 1A) showed a vesicle concentration of 5.73 × 10^8^ particle/mL and a size distribution with a modal value of 113.8 nm, HEC1A showed a vesicle concentration of 9.25 × 10^8^ particle/mL and distribution with a modal value of 130.8 nm, ISHIKAWA showed a vesicle concentration of 3.27 × 10^8^ particle/mL and distribution with a modal value of 120.8 nm, KLE showed a vesicle concentration of 3.27 × 10^8^ particle/mL and distribution with a modal value of 118.4 nm. Then, to check the good performance (Figure 1B) of the EV isolation, we performed Western blotting of common exosome markers CD63 and CD9 and cytosolic proteins in EVs markers HSC70, HSP90 α/β. Altogether, these data suggest that obtained vesicles are enriched in exosomes.

#### Proteomics Study

Then, to characterize the cargo of isolated EVs, we performed a proteomic study. The first step in investigating the proteome of EC cell lines was the bioinformatic processing of data acquired in DDA and DIA. For the DDA data elaboration, we used Proteom Discoverer 3.0. We combined several database search tools (Chimerys, Chimerys+Amanda, and Chimerys+Amanda+Sequest HT) to identify the highest number of proteins. We identified 1813 proteins with Chimerys, 1679 proteins with Chimerys+Amanda+Sequest HT and 1736 proteins with Chimerys+Amanda (Appendix A) with *q*-value < 0.05 and FDR < 1% (Figure 2).

Finally, we have identified 1925 unique proteins by using the three tools. The comparison between the tools showed that Chimerys alone performs better than in combination with other tools. The DIA file was processed with Spectronaut 17. For deep protein identification and quantification, we used the direct DIA (deep) module without libraries. By using this approach, we were able to identify 10,645 proteins with *q* < 0.01 and FDR < 1% (Appendix A). We matched DIA data with DDA data and were able to identify a total of 11,575 different proteins (Figure 3). Analysis showed 995 proteins in common between the two acquisitions, while 9650 proteins in DIA and only 930 in DDA (Appendix A).

The second step was the identification of extracellular matrix (ECM) proteins or matrix-associated proteins. For this purpose, we used the gProfiler tool for ECM protein classification (Figure 4).

gProfiler classified the proteins into groups according to their molecular function, biological processes, and protein class. Regarding molecular functions, proteins were classified mainly in protein binding, catalytic activity, extracellular matrix structural constituent, structural constituent of cytoskeleton, and structural constituent of muscle. Regarding biological processes, proteins were classified into the organonitrogen compound metallic process, intermediate filament cytoskeleton organization, complement activation classical pathway, cortical cytoskeleton organization, and establishment or maintenance of cell polarity. Regarding the cellular component, these were classified as extracellular vesicles, proteosome complex, actin filament bundle, membrane coat, and collagen trimer. Our research with gProfiler, identified 428 proteins as ECM components or related (Appendix A). These EV proteins have been divided according to several biological functions, as reported in Table 1 (*p*-value < 0.05).

We made two comparisons between Type 1 and Type 2 EC cell lines to identify dysregulated proteins (fold change ≥ 1.5 or ≤0.69) (*t*-test *p*-value < 0.05).

The first pairing was between ANCA (Type 1 metastatic)/ISHIKAWA (Type 1 non-metastatic). We identified 49 significant proteins (*p*-value < 0.05) out of which 40 were up-regulated (fold change ≥ 1.5) and 9 down-regulated (fold change ≤ 0.5). Table 2 summarizes the protein abundance trend.

The second pairing was between KLE (metastatic Type 2)/HEC1A (non-metastatic Type 2). In this pairing, we identified 25 significant proteins (*p*-value < 0.05) out of which 8 were up-regulated (fold change ≥ 1.5) and 17 down-regulated (fold change ≤ 0.5). Table 3 summarizes the protein abundance trend.

Reactome is used for the dysregulated protein classification according to their pathways (Figure 5).

Intersecting the proteomic data of the two pairings, we identified seven common proteins: MFGE8, AGRN, POSTN, FBN2, SRPX, PSAP, and PLOD1 (Figure 6). These proteins belong to the surfactant metabolism, molecules associated with elastic fibers pathways.

IPA analysis (Figure 7) showed that these proteins are set in top networks correlated with the ECM corresponding to the following: organization, degradation, binding, remodeling, formation, adhesion of the cell-associated matrix, disassembly, deposition, degeneration, and invasion.

The last step was to compare the proteins identified in our study in the EVs (deep DIA+DDA) with Exocarta and Vesciclepedia databases (Figure 8). We identified 8379 proteins in common, while 2082 were found only in our study and not in databases (Appendix A). All 67 dysregulated proteins identified in both EC cell line comparisons are part of the Exocarta and Vesciclepedia.

### 2.2. Western Blotting

Interestingly, among the differentially regulated proteins, we found four proteins, namely TIMP2, MMP2, MIA, and FLOT1, strongly up-regulate with fold change ≥ 4. Due to its biologically relevant function, we focused on MMP2 (with fold change ≥ 5). A statistically significant higher abundance was found (*t*-test: *p* < 0.05) for MMP2 (*p* = 0.0051) (Figure 9 and Appendix A). In addition, to confirm the relevance of this misregulation also in EC patients, we analyzed MMP2 expression also on 13 tumors from EC patients vs. 13 healthy controls.

In patients, a statistically significant abundance was found (Mann–Whitney sum-rank test: *p* < 0.05) for MMP2 (*p* = 0.04) (Figure 10).

## 3. Discussion

In this study, we applied a deep proteomics approach for the study of the EVs from four Type 1 and 2 metastatic and non-metastatic EC cell lines. Moreover, by Western blotting, we validated MMP2, an extracellular matrix organization protein in AN3CA and ISHIKAWA cell lines.

In this work, we demonstrated that EC cells secrete EVs in bone fide exosomes enriched for ECM components or ECM remodeling proteins.

To investigate this aspect of the tumor microenvironment crosstalk, due to the difficulty of obtaining fresh samples from patients (in particular for type II), we exploited EC cell line models, that indeed allowed us to obtain EVs whose origin was unequivocally from the tumor cells and not from the tumor microenvironment or other tissues. To this aim, we isolated EVs from four different cell lines to model EC type I (Ishikawa, AN3CA) and type 2 (HEC1A, KLE), both of primary and metastatic origin.

A large amount method, based on different principles, has been developed for EVs and exosome isolation, each carrying advantages and disadvantages, as reported by Patel et al. [19] and Brennan K et al. [20]. In this work, we exploited a commercial kit optimized for Exosome isolation. Measurement of vesicle diameter by NTA revealed a population of EVs in the size range of exosomes (116.4 nM in average). Indeed, we also verified the presence of exosomal markers CD9 and CD63 on isolated EVs. Based on these data, we can conclude that the isolated vesicles are in bona fide exosomes.

Next, to characterize purified exosomes and identify cargo proteins, we adopted a deep proteomics approach using DDA and DIA acquisition. Our approach was very efficient and allowed us to identify 11,575 proteins, out of which 2082 were not present in Exocarta and Vesciclepedia.

Proteins are ideal biomolecule predictors of cancer progression, and potential target biomarkers in tumor therapeutics [21].

The use of an AI, such as the Chimerys algorithm for data processing in proteomics, has dramatically improved the identification and quantification of proteins in both DDA and DIA [22].

gProfile analysis revealed that 426 proteins are components of the ECM or associated with the matrix. Reactome analysis has found significant pathways related to the matrix. These pathways are also partly related to metabolism and also fiber formation. Moreover, this showed us that matrix proteins not only modulate ECM-related pathways but are also fundamental in cell metabolism.

IPA analysis gave us a panorama of biofunction interactions by highlighting the organization of the extracellular matrix. ECM is a highly dynamic structure, which is fundamental to maintaining tissue homeostasis [23]. In tumors, the remodeling of ECM is fundamental for promoting tumorigenesis and metastases [24].

Among the misregulated proteins in the EVs, some are very interesting for cancer development and progression according to their function, as briefly summarized here.

MMP2 is a metalloproteinase with key functions in the remodeling of the vasculature, angiogenesis, tissue repair, tumor invasion, and inflammation [25]. The overexpression of the enzyme induces the degradation of the ECM, allowing tumor invasion and metastatization in distant organs [26]. In our patients’ data, Western blotting shows a dysregulation of MMP2. This data is in line with Maity et al. [27] and shows how TIMP2 dysregulation in EC leads to increased MMP2 activity, and this may be related to the modulation of enzyme expression.

Interestingly, a systematic review and meta-analysis of 20 studies confirmed that overexpression of MMP2 is a predictive factor for the poor prognosis of endometrial cancer [28].

Although our data are interesting, further studies need to be performed to correlate levels of MMP2 with age, histotype, grade, tumor size, and stage. To do so, we need to increase our cohort of patients to investigate associations between MMP2 and descriptive and subsequent catamnestic observations.

TIMP2 is the natural inhibitor of metalloproteinases MMP-1, MMP-2, MMP-3, MMP-7, MMP-8, and MMP-9. The protein acts as an irreversible inhibitor binding to their catalytic zinc cofactor [29,30]. Both MMP2 and TIMP2 are considered putative biomarkers of the EC [31]. These proteins have a fundamental role in the development of the disease, in which low levels of MMP2 and high levels of TIMP2 lead to the blocking of metastases. With an increased expression of the MMP2, low expression of TIMP2 was also shown to augment the risk of local and distant metastasis of EC [32]. Rak et al. described a new mechanism of MMP2 in correlation with miR-200b [33]. Their data suggest that microRNA-200b was shown to be overexpressed and implicated in EC by enhancing the activity of MMP2 and downregulating the TIMP2 in the HEC-1A cell line as an in vitro model [34].

FLOT1 is considered as a protein that promotes cancers, ref. [35] influencing the proliferation of cancer cells and epithelial-mesenchymal transition [36]. In EC, Winship et al. have seen an increase in Flot1 in the tumor stroma across grades [37]. FLOT1 behaves as an oncogene in endometrial cancer, leading to growth and tumor invasion by activating regulators like ERK, AKT, and TGF-β [38].

NID1 is a glycoprotein largely present in the basement membrane, closely associated with laminin and other ECMs, like collagen IV and perlecan [39]. Pedrola et al. noted that NID1, cooperating with transcription factor ETV5, leads the cancer cell to a more aggressive profile and metastases [40]. In EC, Ramon et al. showed that inhibition of NID1 and NUPR1 in EC cells with up-regulated ETV5 transcription factor leads to a reduction of tumor growth and migration in vitro and in an orthotopic EC model [41].

TLN1 is overexpressed in several tumors, such as prostate, liver, and oral squamous cell carcinomas [42]. Several studies have shown that TLN1 promotes tumor development and drug resistance in breast cancer [43,44,45]. In an immunohistochemistry study, the authors found that a-actinin and talin cytosynthetic proteins are dysregulated in endometriosis and EC, while ezrin is only in EC. The loss of these proteins, especially in the EC, leads to a change in tissue integrity, resulting in tumor invasion [46].

As in most cancer types, communication between cancer cells and their microenvironment, including stomal cells (fibroblast, macrophages, etc.), promotes tumor growth. In EC, the EVs play a central role in the pathophysiology, enabling the communication between EC cells, cancer-associated fibroblasts (CAFs), and tumor-associated macrophages (TAMs) [47]. EC cell exosomes transfer proteins in CAFs as NEAT1, which is able to modulate the activity of several mi-RNA related to tumor growth in a xenograft model of HEC-1A EC cells [48]. Another interesting communication involved in the development of EC is between EC cells and endometrial stromal cells with tumor-suppressor gene FOXL2 involvement. miRNAs secreted by EC cells, in particular miR-133a, modulate the expression of FOXL2 in endometrial stromal cells, promoting EC progression [49]. TAM infiltration in EC promotes the aggressiveness of cancer [50]. During hypoxia, EC cells produce EVs with immunomodulatory effects. In these Evs, there is an abundance of miR-21, which promotes the proliferation of EC cells via the downregulation of PTEN [51].

In addition to previously reported mechanisms, in this study, we found that EC cells release EVs (in particular exosomes) rich in ECM modifying enzymes and ECM proteins, which might induce ECM remodeling and finally contribute to endometrial carcinogenesis.

## 4. Materials and Methods

### 4.1. Cell Culture

AN3CA (metastatic type 1), HEC1A (non-metastatic type 2), KLE (metastatic type 2), and ISHIKAWA (non-metastatic type 1) are human endometrial cancer cell lines. AN3CA were cultured in Eagle’s Minimum Essential Medium supplemented with 10% FBS (Fetal Bovine Serum); HEC1A were cultured in McCoy’s 5a Medium Modified supplemented with 10% FBS; KLE were cultured in DMEM/Ham’s F12 medium in a 1:1 ratio, with 10% FBS; Ishikawa were cultured with Minimum Essential Medium, supplemented with 2 mM Glutamine + 1% MEM NEAA (Minimum Essential Medium Non-Essential AminoAcids) + 5% FBS. All media were supplemented with penicillin and streptomycin (100 IU/mL each). All the cells described were maintained in a 37 °C, 5% CO_2_ incubator.

AN3CA and HEC1A were obtained from ATCC (Manassas, VA, USA, Cat. n° HTB-111, Cat. n° HTB-112 respectively); KLE were obtained from Istituto Zooprofilattico Sperimentale della Lombardia e Emilia Romagna (Cat. n° BSTCL230), and Ishikawa from CliniSciences (Nanterre, France, Cat. n° ABC-TC1320). Cells are declared free of bacteria and mycoplasmantale by the manufacturer. In addition, cell lines were weekly tested by PCR amplifying DNA sequences coding for conserved regions of the 16S rRNA.

### 4.2. EV Isolation

Cells were seeded and grown in exosome-depleted Fetal Bovine serum (Gibco Thermo Fischer Scientific, Waltham, MA, USA, Cat n° A27208-01) medium for 48 h. Conditioned medium (CM) was collected and centrifuged at 2000× *g* for 30 min to remove cells and debris, if not used immediately, CM was stored at −80 °C. According to the manufacturer protocol, EVs were isolated using total exosome isolation from cell culture media (Invitrogen, Thermo Fischer Scientific, Waltham, MA, USA, Cat. n° 4478359) reagent. For each experimental condition, three biological replicates were prepared and processed in parallel.

### 4.3. Patients

During 2022 and 2023, a total of 26 patients (13 women suffering from EC and 13 non-EC controls) were recruited at the Institute for Maternal and Child Health—IRCCS “Burlo Garofolo” (Trieste, Italy). Our Institute’s Technical and Scientific Committee approved the study, and all procedures complied with the Declaration of Helsinki. All patients signed informed consent forms. The median age of patients (Appendix A) was 72 years (IQR 57–76, Min = 52, Max = 87). The median age of controls (Appendix A) was 42 years (IQR 40–48, Min = 32, Max = 51). Endometrial tissue control samples were derived from hysterectomy performed for ordinary leiomyomas. We excluded patients and controls with human immunodeficiency virus (HIV), hepatitis B virus (HBV), hepatitis C virus (HCV), leiomyoma, and adenomyosis.

### 4.4. Proteome Analysis

A mount of 80 µg of EV proteins, determined previously using Bradford reagent, was digested using EasyPep™ MS Sample Prep Kits (Thermo Fisher). Analysis was performed by nanoflow ultra-high performance liquid chromatography–high-resolution mass spectrometry using an Ultimate 3000 nanoLC (Thermo Fisher Scientific, Bremen, Germany) coupled to an Orbitrap Lumos tribrid mass spectrometer (Thermo Fisher Scientific) with a nanoelectrospray ion source (Thermo Fisher Scientific); 1 μL of the digest was loaded and trapped on a PepMap trap column for 1.00 min at a flow rate of 40 μL/min (Thermo Fisher), and then peptides were separated by a C18 reversed-phase column (250 mm × 75 μm I.D, 2.0 µm, 100 Å, EasySpray, Thermo). Mobile phases were (A) 0.1% HCOOH in H_2_O *v*/*v*, and (B) 0.1% HCOOH in ACN/H_2_O *v*/*v* 80/20, a linear 90 min gradient was performed. Each sample was analyzed in duplicate. The spray voltage was set to 2000 V, while the capillary temperature was 290 °C. MS data were acquired in data-dependent acquisition (DDA), MS1 (range 375–1500 *m*/*z*) was performed at 120,000 resolution, advanced peak determination was used, AGC target and maximum ion injection time were set to standard and auto, respectively, quadrupole isolation was used. MS/MS scan was performed in Orbitrap (OT) at 15,000 resolution. Unassigned precursor ion charge states, as well as singly charged species, were excluded. The isolation window was set to 1.6 Da normalized, and a collision energy (HCD) value of 30 was applied. For MS/MS, the maximum ion injection time for the MS/MS (OT) scan was set to 50 ms, and ACG values were set to standard. The dynamic exclusion was set to 30 s.

For data-independent acquisition (DIA) a first MS scan was performed at 120,000 resolution, DIA was performed in OT at 15,000 resolution, DIA was performed with 10 Dalton isolation windows, AGC target and maximum ion injection time was set to custom using 200 ms and 40 ms values, loop control was set to N = 30. HCD was used with a collision energy value of 30.

The DDA raw data were analyzed using Proteome Discoverer 3.0 with the Sequest HT, AMANDA2.0, and Chimerys-like search engine. We used the following parameter enzyme trypsin, missed cleavages max 2, precursor mass tolerance 10 ppm, and fragment mass tolerance 0.02 Da. Carbamidomethylation was used as a fixed modification, while methionine oxidation was used as the variable. Proteins were considered identified with at least one unique peptide, setting a false discovery rate (FDR) threshold of <1%. The DIA raw data were analyzed using Spectronaut 17. The direct DIA (deep) tool was used for the identification. Carbamidomethylation was used for fix modification, while acetylation and oxidation were used for variable modifications. The FDR was fixed <1%, and proteins were considered to be identified with at least one unique peptide.

### 4.5. Western Blotting

For tissue protein extraction, we used the same procedure as previously described [52]. In these experiments, 100 mg from EC and healthy tissue was homogenized in lysis buffer (1% NP-40, 50 mM Tris-HCl (pH 8.0), NaCl 150 mM) with Phosphatase Inhibitor Cocktail Set II 1× (Millipore, Burlington, VT, USA), 2 mM phenylmethylsulfonyl fluoride (PMSF), and 1 mM benzamidine. Protein concentration was determined by Bradford assay.

Western blotting methodology was conducted as described by [53]. For instance, in Western blotting analysis, 30 µg of both tissue lysate and EVs was loaded onto 4–20% precast gel and then transferred to a nitrocellulose membrane. After protein transfer, the membrane was blocked by treatment with 5% defatted milk in TBS-tween 20 and incubated overnight at 4 °C with 1:1000 diluted primary rabbit polyclonal antibody against MMP2, with antibodies against CD9 (1:800, rabbit polyclonal), CD64 (1:1000, rabbit polyclonal), HSC70 (1:800, rabbit polyclonal), HSP90 α/β (1:800, rabbit polyclonal). After primary antibody incubation, nitrocellulose membranes were washed 3 times with TBS-Tween 0.05% and incubated with HRP-conjugated anti-rabbit IgG (1:3000).

Primary and secondary antibodies were purchased by Sigma-Aldrich (Merck KGaA, Darmstadt, Germany) for the second incubation. Protein band signal visualization was performed by using SuperSignal West Pico Chemiluminescent (Thermo Fisher Scientific Inc., Ottawa, ON, Canada). The intensities of the immunostained bands were normalized with the total protein intensities measured by staining the membranes from the same blot with a Red Ponceau solution (Sigma-Aldrich, St. Louis, MO, USA).

#### Nanoparticle Tracking Analysis

Analysis of concentration and particle size distribution of purified small EVs derived from ISHIKAWA, ANCA3, HEC, and KLE1A were obtained by Nanosight (LM10, Malvern System Ltd., Malvern, UK), equipped with a 405 nm laser. Briefly, each sample was diluted 1:1000 in PBS buffer and recorded for 60 s with a detection threshold set at the maximum. The acquisitions for each sample were made in triple replication.

### 4.6. Bioinformatic Analysis

The gProfiler bioinformatic tool was used for EV protein characterization, according to their molecular function, protein class, and cellular component, while the Reactome tool was used for pathway identification. EnrichVisBox and Morpheus tools were used for bubble plots and heat maps. The Venn diagram tool was used for data correlation in proteomics. The bio-functions were generated via ingenuity pathway analysis (IPA) with a significance of *p* < 0.01, as previously described [54].

### 4.7. Statistical Analysis

Differences were considered significant between EV cell lines type 1 and type 2 when proteins showed a fold change of ±1.5 and satisfied the *t*-test (*p* < 0.05). Differences were considered significant between patients and controls when MMP2 satisfied the Mann–Whitney rank sum test (*p* < 0.05). All analyses were conducted using RStudio and Stata/IC 16.1.

## 5. Conclusions

Our deep proteomics approach allowed us to identify a large number of proteins in EVs in two types of EC. We were able to identify a large number of ECMs and associated proteins and describe several key pathways for tumor development. Our data can open the path to other studies to understand the interaction among cancer cells and the rearrangement of the ECM.

## Figures and Tables

**Figure 1 ijms-25-04650-f001:**
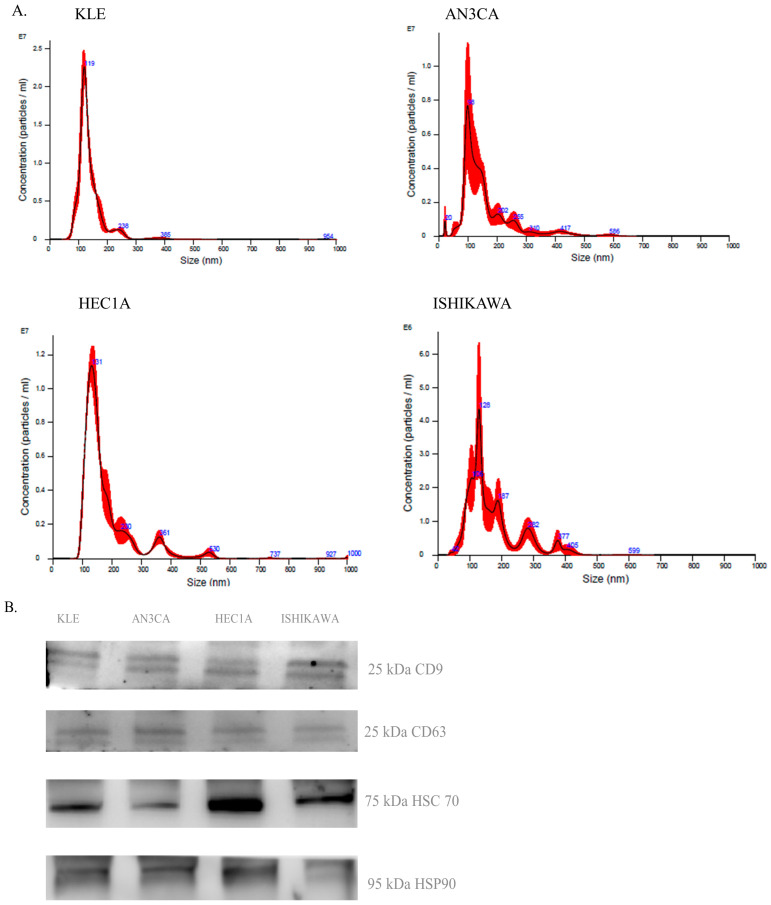
EC cell lines derived EVs characterization. (**A**). Nanoparticle concentration and size distribution of EC EVs obtained through NTA. (**B**). Western blot analysis of vesicle markers (CD63, CD9, HSC70, and HSP 90).

**Figure 2 ijms-25-04650-f002:**
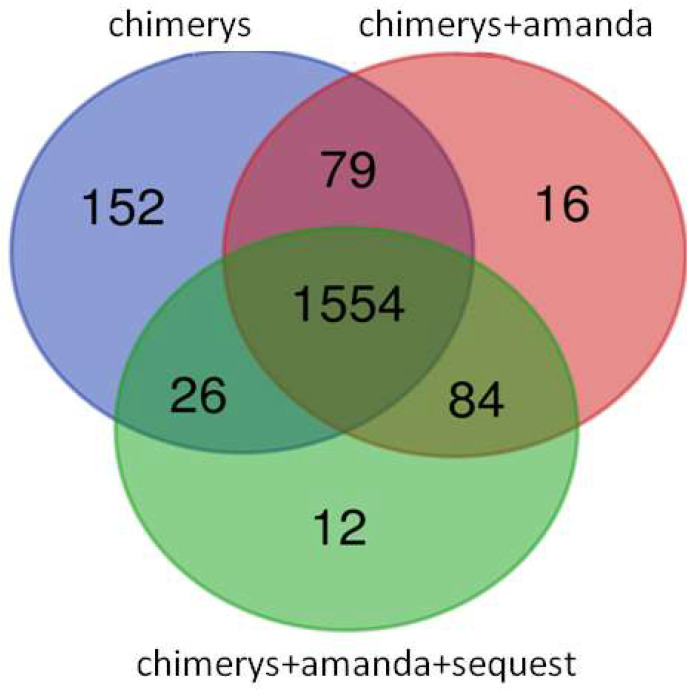
Intersection of identified proteins by three different tools Chimery, Sequest, and Amanda by Venn Diagram. These tools have been combined (Chimerys+Amanda and Chimerys+Amanda +Sequest) to identify as many proteins as possible.

**Figure 3 ijms-25-04650-f003:**
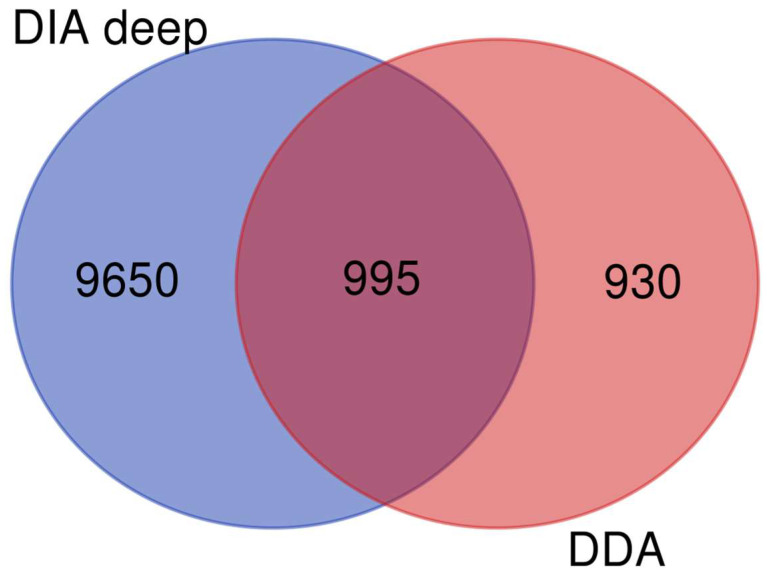
Intersection of DDA with DIA data by Venn Diagram.

**Figure 4 ijms-25-04650-f004:**
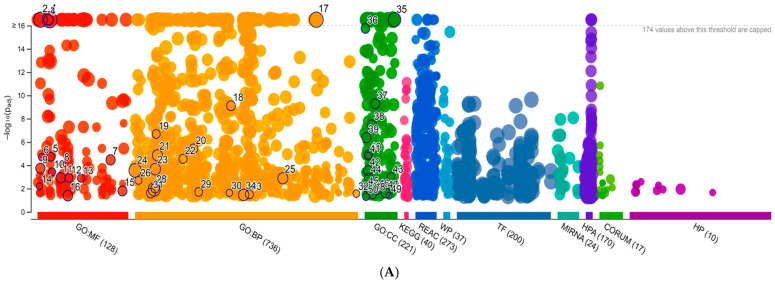
(**A**) An interactive Manhattan plot that illustrates the enrichment analysis results and the (**B**) denominations list of the circles.

**Figure 5 ijms-25-04650-f005:**
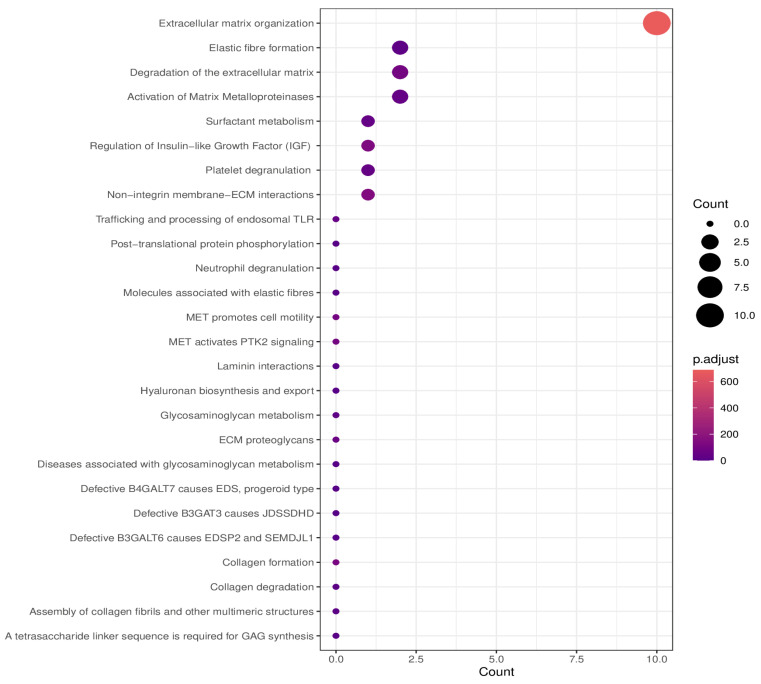
Overview of the proteomic pathways using the EnrichVisBox bioinformatic tool.

**Figure 6 ijms-25-04650-f006:**
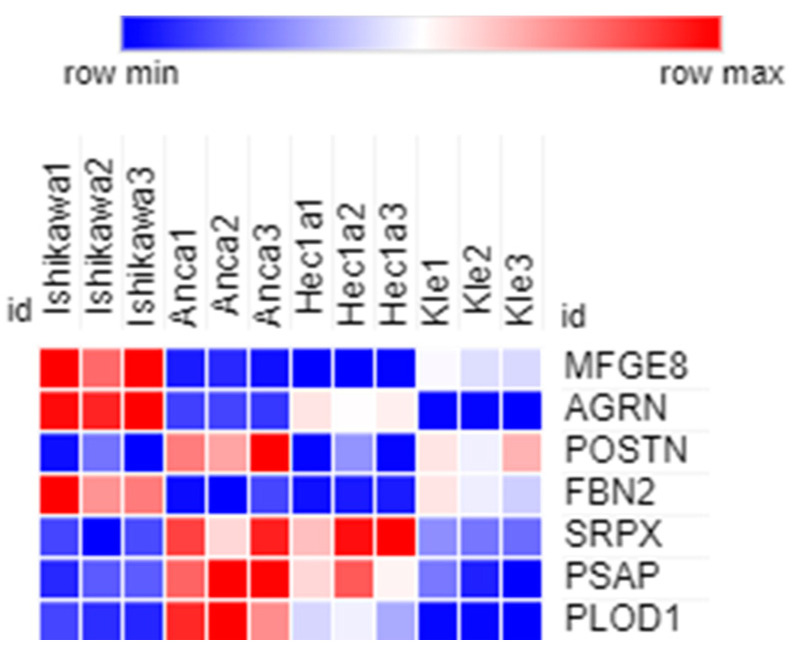
Heat map of the seven common raw data expression proteins using the Morpheus online tool.

**Figure 7 ijms-25-04650-f007:**
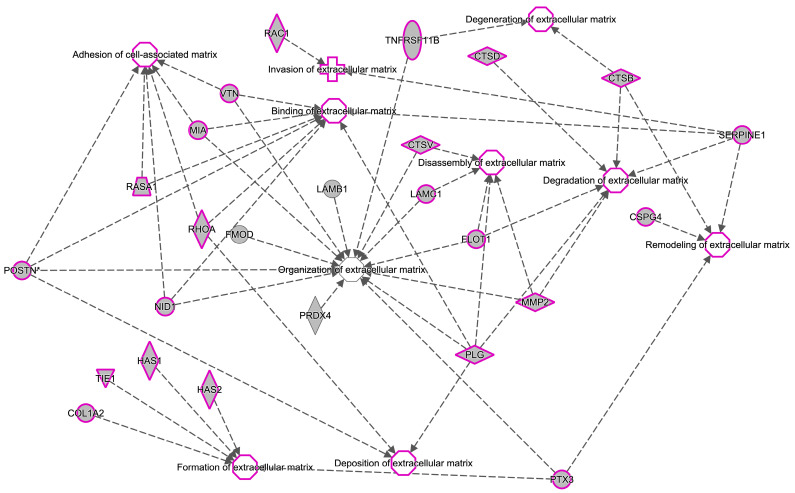
General network build-up from one of the most significant bio-functions.

**Figure 8 ijms-25-04650-f008:**
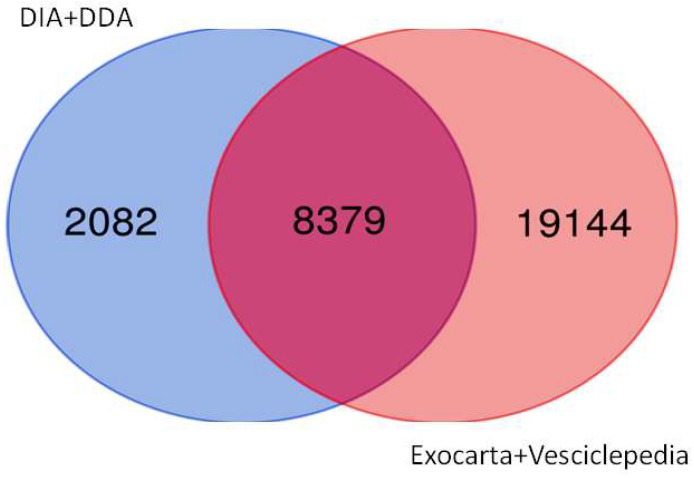
Intersection of DDA+DIA with Exocarta+Vesciclepedia.

**Figure 9 ijms-25-04650-f009:**
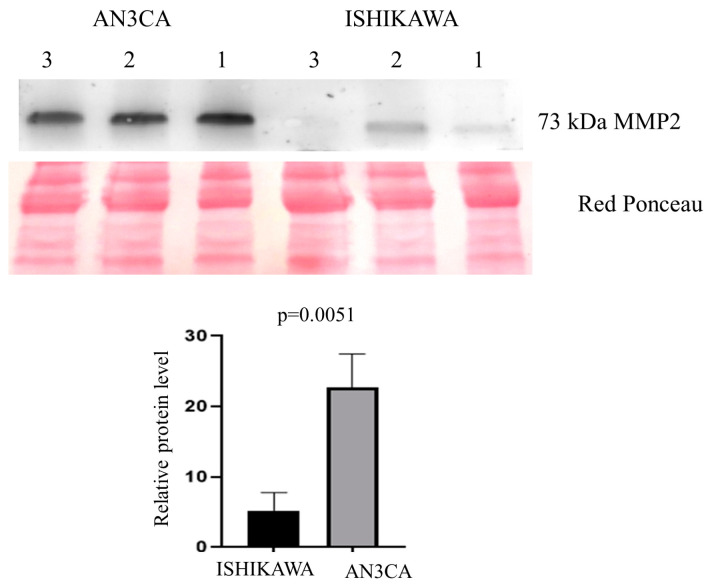
Western blot analysis was utilized to confirm the alteration of protein MMP2 in Ishikawa when compared to AN3CA. The intensity of immunostained bands was normalized against the total protein intensities measured from the same blot stained with Red Ponceau. Results are displayed as a histogram (*p* < 0.05), and each bar represents mean ± standard deviation.

**Figure 10 ijms-25-04650-f010:**
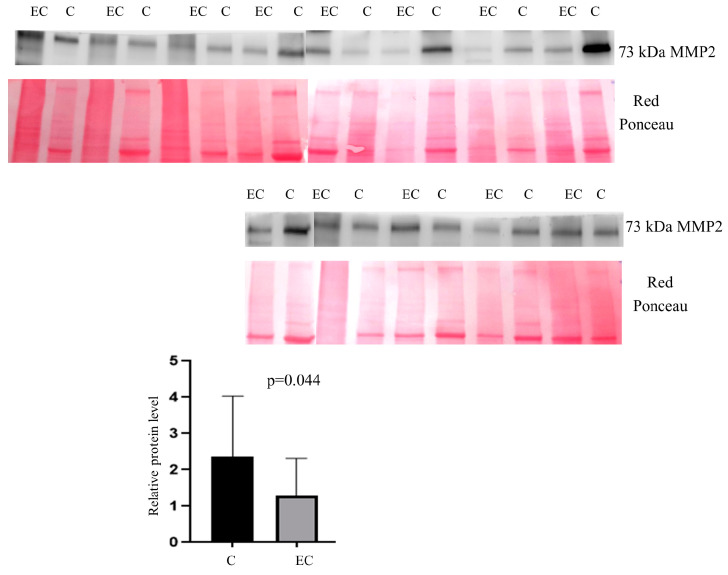
Western blot analysis was utilized to confirm the alteration of protein MMP2 in endometrial cancer (EC) when compared to the normal endometrium (C). The intensity of immunostained bands was normalized against the total protein intensities measured from the same blot stained with Red Ponceau. Results are displayed as a histogram (*p* < 0.05), and each bar represents mean ± standard deviation.

**Table 1 ijms-25-04650-t001:** Biological functions identified by proteomics analysis in EVs, in EC cell types 1 and 2.

Biological Functions	Adjusted *p*-Value
Extracellular matrix structural constituent	4.26 × 10^−24^
Extracellular matrix binding	3.53 × 10^−6^
Extracellular matrix structural constituent conferring tensile strength	4.94 × 10^−5^
Extracellular matrix organization	1.52 × 10^−15^
Cell-matrix adhesion	6.60 × 10^−8^
Regulation of cell-matrix adhesion	2.64 × 10^−5^
Extracellular matrix assembly	2.18 × 10^−3^
Collagen-containing extracellular matrix	1.52 × 10^−43^
Extracellular matrix	2.10 × 10^−42^
Protein complex involved in cell-matrix adhesion	9.51 × 10^−4^
Extracellular matrix organization	4.25 × 10^−16^
Degradation of the extracellular matrix	1.96 × 10^−6^
Cell-extracellular matrix interactions	1.75 × 10^−4^

**Table 2 ijms-25-04650-t002:** Different abundance proteins identified by mass spectrometry present in ANCA/ISHIKAWA paring.

Protein Accessions	Genes	Protein Descriptions	Fold Change	*p* Value
P16035	*TIMP2*	Metalloproteinase inhibitor 2	5.45	0.02
P08253	*MMP2*	72 kDa type IV collagenase	5.002	0.004
Q16674	*MIA*	Melanoma-derived growth regulatory protein	4.26	0.01
O75955	*FLOT1*	Flotillin-1	4.06	0.004
P07858	*CTSB*	Cathepsin B	3.57	0.007
Q06828	*FMOD*	Fibromodulin	3.52	0.02
P10909-2	*CLU*	Isoform 2 of Clusterin	3.5	0.003
P51654	*GPC3*	Glypican-3	3.18	0.012
P04632	*CAPNS1*	Calpain small subunit 1	3.143	0.001
P07602	*PSAP*	Prosaposin	2.76	0.001
Q02809	*PLOD1*	Procollagen-lysine,2-oxoglutarate 5-dioxygenase 1	2.61	0.008
P08123	*COL1A2*	Collagen alpha-2(I) chain	2.61	0.002
Q99467	*CD180*	CD180 antigen	2.47	0.03
O60568	*PLOD3*	Multifunctional procollagen lysine hydroxylase and glycosyltransferase LH3	2.37	0.003
P61586	*RHOA*	Transforming protein RhoA	2.34	0.04
P78539-2	*SRPX*	Isoform 2 of Sushi repeat-containing protein SRPX	2.33	0.01
Q86UW8	*HAPLN4*	Hyaluronan and proteoglycan link protein 4	2.26	0.01
P03950	*ANG*	Angiogenin	2.252	0.03
Q15063	*POSTN*	Periostin	2.1	0.004
Q9GZM7	*TINAGL1*	Tubulointerstitial nephritis antigen-like	2.09	0.03
Q9NVD7	*PARVA*	Alpha-parvin	2.01	0.04
O75325	*LRRN2*	Leucine-rich repeat neuronal protein 2	1.97	0.004
Q9NQX1	*PRDM5*	PR domain zinc finger protein 5	1.96	0.02
P00747	*PLG*	Plasminogen	1.95	0.009
Q9HBI1	*PARVB*	Beta-parvin	1.89	0.0002
P14625	*HSP90B1*	Endoplasmin	1.82	0.03
P07384	*CAPN1*	Calpain-1 catalytic subunit	1.79	0.002
O95967	*EFEMP2*	EGF-containing fibulin-like extracellular matrix protein 2	1.78	0.0005
P63000-2	*RAC1*	Isoform B of Ras-related C3 botulinum toxin substrate 1	1.78	0.03
P27797	*CALR*	Calreticulin	1.74	0.04
P35590	*TIE1*	Tyrosine-protein kinase receptor Tie-1	1.74	0.01
Q6UVK1	*CSPG4*	Chondroitin sulfate proteoglycan 4	1.71	0.02
Q13162	*PRDX4*	Peroxiredoxin-4	1.64	0.001
P35858	*IGFALS*	Insulin-like growth factor-binding protein complex acid labile subunit	1.64	0.01
Q8NI99	*ANGPTL6*	Angiopoietin-related protein 6	1.62	0.01
P04004	*VTN*	Vitronectin	1.62	0.01
Q13751	*LAMB3*	Laminin subunit beta-3	1.58	0.01
Q92819	*HAS2*	Hyaluronan synthase 2	1.57	0.02
P19823	*ITIH2*	Inter-alpha-trypsin inhibitor heavy chain H2	1.54	0.03
P35555	*FBN1*	Fibrillin-1	1.5	0.01
P07942	*LAMB1*	Laminin subunit beta-1	0.62	0.004
Q92839	*HAS1*	Hyaluronan synthase 1	0.59	0.01
P11047	*LAMC1*	Laminin subunit gamma-1	0.59	0.003
P20936	*RASA1*	Ras GTPase-activating protein 1	0.58	0.01
Q96A84-2	*EMID1*	Isoform 2 of EMI domain-containing protein 1	0.55	0.006
O15230	*LAMA5*	Laminin subunit alpha-5	0.36	0.006
P35556	*FBN2*	Fibrillin-2	0.32	0.005
O00468-6	*AGRN*	Isoform 6 of Agrin	0.20	0.0002
Q08431	*MFGE8*	Lactadherin	0.12	0.005

**Table 3 ijms-25-04650-t003:** Different abundance proteins identified by mass spectrometry present in KLE/HEC1A paring.

Protein Accessions	Genes	Protein Descriptions	Fold Change	*p* Value
P14543	*NID1*	Nidogen-1	14.53	1.48 × 10^−6^
Q08431	*MFGE8*	Lactadherin	8.3	0.001
Q9Y490	*TLN1*	Talin-1	2.17	0.003
P35556	*FBN2*	Fibrillin-2	2.13	0.009
Q9ULV4-3	*CORO1C*	Isoform 3 of coronin-1C	1.92	0.04
O60911	*CTSV*	Cathepsin L2	1.73	0.03
Q15063	*POSTN*	Periostin	1.66	0.01
Q92743	*HTRA1*	Serine protease HTRA1	1.56	0.003
Q6NUI6	*CHADL*	Chondroadherin-like protein	0.64	0.01
P26022	*PTX3*	Pentraxin-related protein PTX3	0.61	0.01
P80108	*GPLD1*	Phosphatidylinositol-glycan-specific phospholipase D	0.6	0.008
P07355-2	*ANXA2*	Isoform 2 of Annexin A2	0.59	0.009
P26447	*S100A4*	Protein S100-A4	0.57	0.0005
Q7RTU9	*STRC*	Stereocilin	0.54	0.04
P78539-2	*SRPX*	Isoform 2 of sushi repeat-containing protein SRPX	0.51	0.03
Q02809	*PLOD1*	Procollagen-lysine,2-oxoglutarate 5-dioxygenase 1	0.5	0.008
P07339	*CTSD*	Cathepsin D	0.5	0.005
P05121	*SERPINE1*	Plasminogen activator inhibitor 1	0.48	0.02
P07602	*PSAP*	Prosaposin	0.42	0.01
Q9UPN3	*MACF1*	Microtubule-actin cross-linking factor 1, isoforms 1/2/3/4/5	0.41	0.01
P53634	*CTSC*	Dipeptidyl peptidase 1	0.31	0.01
Q08380	*LGALS3BP*	Galectin-3-binding protein	0.3	0.003
O00300	*TNFRSF11B*	Tumor necrosis factor receptor superfamily member 11B	0.18	0.001
O00468-6	*AGRN*	Isoform 6 of agrin	0.16	9.41 × 10^−5^
Q16270	*IGFBP7*	Insulin-like growth factor-binding protein 7	0.12	0.03

## Data Availability

The data presented in this study are available on request from the corresponding author. The data are not publicly available due to ethical reasons.

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
