# Peer review of "The Deep Proteomics Approach Identified Extracellular Vesicular Proteins Correlated to Extracellular Matrix in Type One and Two Endometrial Cancer"

_ijms, 2024, doi:10.3390/ijms25094650_

Round 1
Reviewer 1 Report
Comments and Suggestions for Authors
The aim of the results described in this manuscript was, according to authors, “to characterize the EV proteome, in order to identify dysregulated ECMs and associated proteins’. Actually, it is not clear why the authors, having information on the composition of the entire EVs proteome, focused only on ECM and ECM-related proteins. Progression of any cancer, including endometrial cancer, is accompanied by remodeling of the tumor niche. But in order to talk about the possibility of distinguishing type 1 from type 2 endometrial cancer, including metastatic or non-metastatic, in my opinion, it is worth identifying all proteins , the amount of which is changed, and only on this basis infer the changes occurring at the molecular level in these cells. Manuscript is hard to read. At this stage, it gives the impression of a draft.
Comments:
1. Lines 5-7: The numbers next to the names indicating the corresponding affiliation should be placed in the superscript.
2. Line 6: Please add * next to the appropriate name (Monasta).
3. Line 23: Please use another symbol to mark the equal contributions of two authors (V. Capaci and F. Kharrat) and add this mark next to these names in line 5.
4. Lines 25-26 and line 49: In the subject literature, I have not encountered the term that EVs are cellular fragments. Please use a different term. As far as their biogenesis is concerned, cell membrane blubbing is only one of three possible ways for EVs to form. Besides, EV cargo contains a much richer set of molecules than just proteins involved in immune response and apoptosis, as the description in the Abstract suggests.
5. Lines 29: From the description in the methodology, it is known that EVs were isolated from the conditioned medium obtained from the culture of 4 human endometrial cancer cell lines. Thus, the term "four EV cell lines" is incorrect.
6. Lines 35: The abbreviation EC, which was not previously defined, was introduced here. Please define it. My guess is that it refers to endometrial cancer. But it should be clarified.
7. Lines 45-48: No brief characterization of type 2 endometrial cancer analogous to type 1.
8. Line 76: CD63 and CD9 are not markers of EVs, but of one of their subpopulations, that is, exosomes.
9. Figure 2: According to lines 82-83 in Figure 2, there should be groups: Chimerys, 82 Chimerys+Sequest HT and Chimerys+Amanda+Sequest HT. There should not be Chimerys+Amanda.
10. Lines 84-86: To verify that the protein numbers given are correct. Based on the information in Figure 2, other numbers come out of the addition. The ones in parentheses: We identified 1813 (1811) proteins with Chimerys , 1679 (1676)proteins 84 with Chimerys+Amanda+Sequest HT and 1736 (1733) proteins with Chimerys+Amanda (Sup-85 plementary Table S1) with q-value < 0.05 and FDR < 1% (Figure 2).
11. Lines 100-102 and 106-107: These two sentences contradict each other. After all, what was gProfiler used for, to identify proteins or to classify them.
12. Figure 4: Too small font makes this figure poorly readable.
13. Lines 116-117: Why were the results for the molecular function and cellular component not also presented in table form, analogous to what was done for the biological function in Table 1.
14. Table 2 and 3: What is the justification for specifying Fold Change over P Value with such a large number of decimal places?
15. Line 189: Once again, adding up the numbers in Figure 8 comes out with a different number than 11575.
16. Lines 175-229: The Discussion section looks like a script for writing a proper Discussion. Brief information about 7 proteins, actually not sure why selected. You have to look only in Tables 2 and 3 that these are the proteins that had the biggest fold change. And any link between these proteins and endometrial cancer is missing.
17. Lines 231-242: No information is available on which of the cell lines used in the study represents type 1 and which represents type 2 endometrial cancer and metastatic or non-metastatic.
18. Line 239: CO2 instead of CO2 (2 in subscript).
19. Lines 243-249: According to ISEV recommendations, characterization of isolated EV samples should also include transmission electron microscopy (TEM) and nanoparticle tracking analysis (NTA).
20. Lines 271: H2O instead of H2O (2 in subscript).
21. Lines 305: EVs instead of EV.
22. Lines 307-308: Unnecessary repetitions: incubated overnight at 4 °C with incubated overnight at 4 °C with 1:1000 diluted primary
23. Lines 309-310: No information about the companies from which the used antibodies were purchased. No information on the dilution of anti-MMP2 antibodies is available.
24. Lines 314-317: I have not yet encountered such a way of normalization. Rather, reference proteins, such as actin, are used in such a case.
Comments on the Quality of English LanguageModerate editing of English language required.
Author Response
Reviewer 1
Lines 5-7: The numbers next to the names indicating the corresponding affiliation should be placed in the superscript.
Our reply: We fixed the corresponding affiliation.
Line 6: Please add * next to the appropriate name (Monasta).
Our reply: We fixed the name Monasta.
Line 23: Please use another symbol to mark the equal contributions of two authors (V. Capaci and F. Kharrat) and add this mark next to these names in line 5.
Our reply: We fixed the names.
Lines 25-26 and line 49: In the subject literature, I have not encountered the term that EVs are cellular fragments. Please use a different term. As far as their biogenesis is concerned, cell membrane blubbing is only one of three possible ways for EVs to form. Besides, EV cargo contains a much richer set of molecules than just proteins involved in immune response and apoptosis, as the description in the Abstract suggests.
Our reply: We removed the term cellular fragments. This term was replaced and lines 25 and line 49 were fixed.
Lines 29: From the description in the methodology, it is known that EVs were isolated from the conditioned medium obtained from the culture of 4 human endometrial cancer cell lines. Thus, the term "four EV cell lines" is incorrect.
Our reply: We fixed the line 29.
Lines 35: The abbreviation EC, which was not previously defined, was introduced here. Please define it. My guess is that it refers to endometrial cancer. But it should be clarified.
Our reply: The abbreviation EC correspond to endometrial cancer. We fixed it.
Lines 45-48: No brief characterization of type 2 endometrial cancer analogous to type 1.
Our reply: We added characterization of type 2 endometrial cancer in the introduction.
Line 76: CD63 and CD9 are not markers of EVs, but of one of their subpopulations, that is, exosomes.
Our reply: We added new paragraphs describing the characterization of EVs.
.
Figure 2: According to lines 82-83 in Figure 2, there should be groups: Chimerys, 82 Chimerys+Sequest HT and Chimerys+Amanda+Sequest HT. There should not be Chimerys+Amanda.
Our reply: We fixed the lines 82-83.
Lines 100-102 and 106-107: These two sentences contradict each other. After all, what was gProfiler used for, to identify proteins or to classify them.
Our reply: We fixed the lines. gProfiler is used to classify proteins.
Figure 4: Too small font makes this figure poorly readable.
Our reply: We fixed the figure 4
Lines 116-117: Why were the results for the molecular function and cellular component not also presented in table form, analogous to what was done for the biological function in Table 1.
Our reply: The EV proteins have been divided according to several biological functions not in biological processes. We fixed this error on table 1.
Table 2 and 3: What is the justification for specifying Fold Change over P Value with such a large number of decimal places?
Our reply: This is given by the analyses made with RStudio. We fixed the decimals number.
Line 189: Once again, adding up the numbers in Figure 8 comes out with a different number than 11575.
Our reply: In this case the number 11575 corresponds to the total accession number identified in the study. In the case of Figure 8, the names of genes were used to compare DIA+DDA with Exocarta and Vesciclepedia databases by omitting information. Unfortunately, the two databases are based on the name of the gene, not the accession number.
Lines 175-229: The Discussion section looks like a script for writing a proper Discussion. Brief information about 7 proteins, actually not sure why selected. You have to look only in Tables 2 and 3 that these are the proteins that had the biggest fold change. And any link between these proteins and endometrial cancer is missing.
Our reply: We added in discussion the link between these proteins and endometrial cancer.
Lines 231-242: No information is available on which of the cell lines used in the study represents type 1 and which represents type 2 endometrial cancer and metastatic or non-metastatic.
Our reply: We added information on the cell lines.
Line 239: CO2 instead of CO2 (2 in subscript).
Our reply: We fixed it.
Lines 243-249: According to ISEV recommendations, characterization of isolated EV samples should also include transmission electron microscopy (TEM) and nanoparticle tracking analysis (NTA).
Our reply: According to ISEV recommendations, we have conducted characterization of isolated EV by nanoparticle tracking analysis (NTA). We measured the concentration and diameter of the EVs. Although the TEM measures would have improved the analysis unfortunately we do not have access to the instrument.
Lines 271: H2O instead of H2O (2 in subscript).
Our reply: We fixed it.
Lines 305: EVs instead of EV.
Our reply: We fixed it.
Lines 307-308: Unnecessary repetitions: incubated overnight at 4 °C with incubated overnight at 4 °C with 1:1000 diluted primary.
Our reply: We fixed it.
Lines 309-310: No information about the companies from which the used antibodies were purchased. No information on the dilution of anti-MMP2 antibodies is available.
Our reply: We added the company from which the used antibodies were purchased. The anti-MMP2 was used 1:1000.
I have not yet encountered such a way of normalization. Rather, reference proteins, such as actin, are used in such a case.
Our reply: To normalize the results of the WB analysis, we decided to determine the total protein content of each sample by Red Ponceau, since increasing data in the literature reported that total protein normalization outperform housekeeping protein immunodetection as loading controls (B. Rivero-Gutiérrez, A. Anzola, O. Martínez-Augustin, F. Sánchez de Medina. Stain-free detection as loading control alternative to Ponceau and housekeeping protein immunodetection in Western blotting, Analytical Biochemistry, Volume 467, 2014,) (Sander H, Wallace S, Plouse R, Tiwari S, Gomes AV. Ponceau S waste: Ponceau S staining for total protein normalization. Anal Biochem. 2019 Jun 15;575:44-53.). Moreover, in EC most of the proteins commonly used as housekeeping are indeed affected and thus not adequate to be used as a control for normalization. For example, according to our previous published works, GAPDH is up-regulated in EC (Ura B, Monasta L, Arrigoni G, Franchin C, Radillo O, Peterlunger I, Ricci G, Scrimin F: A proteomic approach for the identification of biomarkers in endometrial cancer uterine aspirate. Oncotarget. 2017 Dec 12; 8: 109536–109545), while ACTB is down-regulated in the serum of EC patients (Ura B, Biffi S, Monasta L, Arrigoni G, Battisti I, Di Lorenzo G, Federico Romano F, Aloisio M, Celsi F, Addobbati R, Valle F, RampazzoE , Brucale M, ARidolfi A, Licastro D, Ricci G: Two Dimensional-Difference in Gel Electrophoresis (2D-DIGE) Proteomic Approach for the Identification of Biomarkers in Endometrial Cancer Serum. Cancers 2021, 13, 3639). Similarly, tubulin has also been found up-regulated in EC (Charles Dumontet; Mary Ann Jordan; Francis F.Y. Lee. Ixabepilone: targeting βIII-tubulin expression in taxane-resistant malignancies. Mol Cancer Ther (2009) 8 (1): 17–25). We thus decided to apply a total protein content normalization method, because we could not establish which proteins should be considered as good housekeeping in our samples.
Reviewer 1
Lines 5-7: The numbers next to the names indicating the corresponding affiliation should be placed in the superscript.
Our reply: We fixed the corresponding affiliation.
Line 6: Please add * next to the appropriate name (Monasta).
Our reply: We fixed the name Monasta.
Line 23: Please use another symbol to mark the equal contributions of two authors (V. Capaci and F. Kharrat) and add this mark next to these names in line 5.
Our reply: We fixed the names.
Lines 25-26 and line 49: In the subject literature, I have not encountered the term that EVs are cellular fragments. Please use a different term. As far as their biogenesis is concerned, cell membrane blubbing is only one of three possible ways for EVs to form. Besides, EV cargo contains a much richer set of molecules than just proteins involved in immune response and apoptosis, as the description in the Abstract suggests.
Our reply: We removed the term cellular fragments. This term was replaced and lines 25 and line 49 were fixed.
Lines 29: From the description in the methodology, it is known that EVs were isolated from the conditioned medium obtained from the culture of 4 human endometrial cancer cell lines. Thus, the term "four EV cell lines" is incorrect.
Our reply: We fixed the line 29.
Lines 35: The abbreviation EC, which was not previously defined, was introduced here. Please define it. My guess is that it refers to endometrial cancer. But it should be clarified.
Our reply: The abbreviation EC correspond to endometrial cancer. We fixed it.
Lines 45-48: No brief characterization of type 2 endometrial cancer analogous to type 1.
Our reply: We added characterization of type 2 endometrial cancer in the introduction.
Line 76: CD63 and CD9 are not markers of EVs, but of one of their subpopulations, that is, exosomes.
Our reply: We added new paragraphs describing the characterization of EVs.
.
Figure 2: According to lines 82-83 in Figure 2, there should be groups: Chimerys, 82 Chimerys+Sequest HT and Chimerys+Amanda+Sequest HT. There should not be Chimerys+Amanda.
Our reply: We fixed the lines 82-83.
Lines 100-102 and 106-107: These two sentences contradict each other. After all, what was gProfiler used for, to identify proteins or to classify them.
Our reply: We fixed the lines. gProfiler is used to classify proteins.
Figure 4: Too small font makes this figure poorly readable.
Our reply: We fixed the figure 4
Lines 116-117: Why were the results for the molecular function and cellular component not also presented in table form, analogous to what was done for the biological function in Table 1.
Our reply: The EV proteins have been divided according to several biological functions not in biological processes. We fixed this error on table 1.
Table 2 and 3: What is the justification for specifying Fold Change over P Value with such a large number of decimal places?
Our reply: This is given by the analyses made with RStudio. We fixed the decimals number.
Line 189: Once again, adding up the numbers in Figure 8 comes out with a different number than 11575.
Our reply: In this case the number 11575 corresponds to the total accession number identified in the study. In the case of Figure 8, the names of genes were used to compare DIA+DDA with Exocarta and Vesciclepedia databases by omitting information. Unfortunately, the two databases are based on the name of the gene, not the accession number.
Lines 175-229: The Discussion section looks like a script for writing a proper Discussion. Brief information about 7 proteins, actually not sure why selected. You have to look only in Tables 2 and 3 that these are the proteins that had the biggest fold change. And any link between these proteins and endometrial cancer is missing.
Our reply: We added in discussion the link between these proteins and endometrial cancer.
Lines 231-242: No information is available on which of the cell lines used in the study represents type 1 and which represents type 2 endometrial cancer and metastatic or non-metastatic.
Our reply: We added information on the cell lines.
Line 239: CO2 instead of CO2 (2 in subscript).
Our reply: We fixed it.
Lines 243-249: According to ISEV recommendations, characterization of isolated EV samples should also include transmission electron microscopy (TEM) and nanoparticle tracking analysis (NTA).
Our reply: According to ISEV recommendations, we have conducted characterization of isolated EV by nanoparticle tracking analysis (NTA). We measured the concentration and diameter of the EVs. Although the TEM measures would have improved the analysis unfortunately we do not have access to the instrument.
Lines 271: H2O instead of H2O (2 in subscript).
Our reply: We fixed it.
Lines 305: EVs instead of EV.
Our reply: We fixed it.
Lines 307-308: Unnecessary repetitions: incubated overnight at 4 °C with incubated overnight at 4 °C with 1:1000 diluted primary.
Our reply: We fixed it.
Lines 309-310: No information about the companies from which the used antibodies were purchased. No information on the dilution of anti-MMP2 antibodies is available.
Our reply: We added the company from which the used antibodies were purchased. The anti-MMP2 was used 1:1000.
I have not yet encountered such a way of normalization. Rather, reference proteins, such as actin, are used in such a case.
Our reply: To normalize the results of the WB analysis, we decided to determine the total protein content of each sample by Red Ponceau, since increasing data in the literature reported that total protein normalization outperform housekeeping protein immunodetection as loading controls (B. Rivero-Gutiérrez, A. Anzola, O. Martínez-Augustin, F. Sánchez de Medina. Stain-free detection as loading control alternative to Ponceau and housekeeping protein immunodetection in Western blotting, Analytical Biochemistry, Volume 467, 2014,) (Sander H, Wallace S, Plouse R, Tiwari S, Gomes AV. Ponceau S waste: Ponceau S staining for total protein normalization. Anal Biochem. 2019 Jun 15;575:44-53.). Moreover, in EC most of the proteins commonly used as housekeeping are indeed affected and thus not adequate to be used as a control for normalization. For example, according to our previous published works, GAPDH is up-regulated in EC (Ura B, Monasta L, Arrigoni G, Franchin C, Radillo O, Peterlunger I, Ricci G, Scrimin F: A proteomic approach for the identification of biomarkers in endometrial cancer uterine aspirate. Oncotarget. 2017 Dec 12; 8: 109536–109545), while ACTB is down-regulated in the serum of EC patients (Ura B, Biffi S, Monasta L, Arrigoni G, Battisti I, Di Lorenzo G, Federico Romano F, Aloisio M, Celsi F, Addobbati R, Valle F, RampazzoE , Brucale M, ARidolfi A, Licastro D, Ricci G: Two Dimensional-Difference in Gel Electrophoresis (2D-DIGE) Proteomic Approach for the Identification of Biomarkers in Endometrial Cancer Serum. Cancers 2021, 13, 3639). Similarly, tubulin has also been found up-regulated in EC (Charles Dumontet; Mary Ann Jordan; Francis F.Y. Lee. Ixabepilone: targeting βIII-tubulin expression in taxane-resistant malignancies. Mol Cancer Ther (2009) 8 (1): 17–25). We thus decided to apply a total protein content normalization method, because we could not establish which proteins should be considered as good housekeeping in our samples.
Reviewer 1
Lines 5-7: The numbers next to the names indicating the corresponding affiliation should be placed in the superscript.
Our reply: We fixed the corresponding affiliation.
Line 6: Please add * next to the appropriate name (Monasta).
Our reply: We fixed the name Monasta.
Line 23: Please use another symbol to mark the equal contributions of two authors (V. Capaci and F. Kharrat) and add this mark next to these names in line 5.
Our reply: We fixed the names.
Lines 25-26 and line 49: In the subject literature, I have not encountered the term that EVs are cellular fragments. Please use a different term. As far as their biogenesis is concerned, cell membrane blubbing is only one of three possible ways for EVs to form. Besides, EV cargo contains a much richer set of molecules than just proteins involved in immune response and apoptosis, as the description in the Abstract suggests.
Our reply: We removed the term cellular fragments. This term was replaced and lines 25 and line 49 were fixed.
Lines 29: From the description in the methodology, it is known that EVs were isolated from the conditioned medium obtained from the culture of 4 human endometrial cancer cell lines. Thus, the term "four EV cell lines" is incorrect.
Our reply: We fixed the line 29.
Lines 35: The abbreviation EC, which was not previously defined, was introduced here. Please define it. My guess is that it refers to endometrial cancer. But it should be clarified.
Our reply: The abbreviation EC correspond to endometrial cancer. We fixed it.
Lines 45-48: No brief characterization of type 2 endometrial cancer analogous to type 1.
Our reply: We added characterization of type 2 endometrial cancer in the introduction.
Line 76: CD63 and CD9 are not markers of EVs, but of one of their subpopulations, that is, exosomes.
Our reply: We added new paragraphs describing the characterization of EVs.
.
Figure 2: According to lines 82-83 in Figure 2, there should be groups: Chimerys, 82 Chimerys+Sequest HT and Chimerys+Amanda+Sequest HT. There should not be Chimerys+Amanda.
Our reply: We fixed the lines 82-83.
Lines 100-102 and 106-107: These two sentences contradict each other. After all, what was gProfiler used for, to identify proteins or to classify them.
Our reply: We fixed the lines. gProfiler is used to classify proteins.
Figure 4: Too small font makes this figure poorly readable.
Our reply: We fixed the figure 4
Lines 116-117: Why were the results for the molecular function and cellular component not also presented in table form, analogous to what was done for the biological function in Table 1.
Our reply: The EV proteins have been divided according to several biological functions not in biological processes. We fixed this error on table 1.
Table 2 and 3: What is the justification for specifying Fold Change over P Value with such a large number of decimal places?
Our reply: This is given by the analyses made with RStudio. We fixed the decimals number.
Line 189: Once again, adding up the numbers in Figure 8 comes out with a different number than 11575.
Our reply: In this case the number 11575 corresponds to the total accession number identified in the study. In the case of Figure 8, the names of genes were used to compare DIA+DDA with Exocarta and Vesciclepedia databases by omitting information. Unfortunately, the two databases are based on the name of the gene, not the accession number.
Lines 175-229: The Discussion section looks like a script for writing a proper Discussion. Brief information about 7 proteins, actually not sure why selected. You have to look only in Tables 2 and 3 that these are the proteins that had the biggest fold change. And any link between these proteins and endometrial cancer is missing.
Our reply: We added in discussion the link between these proteins and endometrial cancer.
Lines 231-242: No information is available on which of the cell lines used in the study represents type 1 and which represents type 2 endometrial cancer and metastatic or non-metastatic.
Our reply: We added information on the cell lines.
Line 239: CO2 instead of CO2 (2 in subscript).
Our reply: We fixed it.
Lines 243-249: According to ISEV recommendations, characterization of isolated EV samples should also include transmission electron microscopy (TEM) and nanoparticle tracking analysis (NTA).
Our reply: According to ISEV recommendations, we have conducted characterization of isolated EV by nanoparticle tracking analysis (NTA). We measured the concentration and diameter of the EVs. Although the TEM measures would have improved the analysis unfortunately we do not have access to the instrument.
Lines 271: H2O instead of H2O (2 in subscript).
Our reply: We fixed it.
Lines 305: EVs instead of EV.
Our reply: We fixed it.
Lines 307-308: Unnecessary repetitions: incubated overnight at 4 °C with incubated overnight at 4 °C with 1:1000 diluted primary.
Our reply: We fixed it.
Lines 309-310: No information about the companies from which the used antibodies were purchased. No information on the dilution of anti-MMP2 antibodies is available.
Our reply: We added the company from which the used antibodies were purchased. The anti-MMP2 was used 1:1000.
I have not yet encountered such a way of normalization. Rather, reference proteins, such as actin, are used in such a case.
Our reply: To normalize the results of the WB analysis, we decided to determine the total protein content of each sample by Red Ponceau, since increasing data in the literature reported that total protein normalization outperform housekeeping protein immunodetection as loading controls (B. Rivero-Gutiérrez, A. Anzola, O. Martínez-Augustin, F. Sánchez de Medina. Stain-free detection as loading control alternative to Ponceau and housekeeping protein immunodetection in Western blotting, Analytical Biochemistry, Volume 467, 2014,) (Sander H, Wallace S, Plouse R, Tiwari S, Gomes AV. Ponceau S waste: Ponceau S staining for total protein normalization. Anal Biochem. 2019 Jun 15;575:44-53.). Moreover, in EC most of the proteins commonly used as housekeeping are indeed affected and thus not adequate to be used as a control for normalization. For example, according to our previous published works, GAPDH is up-regulated in EC (Ura B, Monasta L, Arrigoni G, Franchin C, Radillo O, Peterlunger I, Ricci G, Scrimin F: A proteomic approach for the identification of biomarkers in endometrial cancer uterine aspirate. Oncotarget. 2017 Dec 12; 8: 109536–109545), while ACTB is down-regulated in the serum of EC patients (Ura B, Biffi S, Monasta L, Arrigoni G, Battisti I, Di Lorenzo G, Federico Romano F, Aloisio M, Celsi F, Addobbati R, Valle F, RampazzoE , Brucale M, ARidolfi A, Licastro D, Ricci G: Two Dimensional-Difference in Gel Electrophoresis (2D-DIGE) Proteomic Approach for the Identification of Biomarkers in Endometrial Cancer Serum. Cancers 2021, 13, 3639). Similarly, tubulin has also been found up-regulated in EC (Charles Dumontet; Mary Ann Jordan; Francis F.Y. Lee. Ixabepilone: targeting βIII-tubulin expression in taxane-resistant malignancies. Mol Cancer Ther (2009) 8 (1): 17–25). We thus decided to apply a total protein content normalization method, because we could not establish which proteins should be considered as good housekeeping in our samples.
Reviewer 1
Lines 5-7: The numbers next to the names indicating the corresponding affiliation should be placed in the superscript.
Our reply: We fixed the corresponding affiliation.
Line 6: Please add * next to the appropriate name (Monasta).
Our reply: We fixed the name Monasta.
Line 23: Please use another symbol to mark the equal contributions of two authors (V. Capaci and F. Kharrat) and add this mark next to these names in line 5.
Our reply: We fixed the names.
Lines 25-26 and line 49: In the subject literature, I have not encountered the term that EVs are cellular fragments. Please use a different term. As far as their biogenesis is concerned, cell membrane blubbing is only one of three possible ways for EVs to form. Besides, EV cargo contains a much richer set of molecules than just proteins involved in immune response and apoptosis, as the description in the Abstract suggests.
Our reply: We removed the term cellular fragments. This term was replaced and lines 25 and line 49 were fixed.
Lines 29: From the description in the methodology, it is known that EVs were isolated from the conditioned medium obtained from the culture of 4 human endometrial cancer cell lines. Thus, the term "four EV cell lines" is incorrect.
Our reply: We fixed the line 29.
Lines 35: The abbreviation EC, which was not previously defined, was introduced here. Please define it. My guess is that it refers to endometrial cancer. But it should be clarified.
Our reply: The abbreviation EC correspond to endometrial cancer. We fixed it.
Lines 45-48: No brief characterization of type 2 endometrial cancer analogous to type 1.
Our reply: We added characterization of type 2 endometrial cancer in the introduction.
Line 76: CD63 and CD9 are not markers of EVs, but of one of their subpopulations, that is, exosomes.
Our reply: We added new paragraphs describing the characterization of EVs.
.
Figure 2: According to lines 82-83 in Figure 2, there should be groups: Chimerys, 82 Chimerys+Sequest HT and Chimerys+Amanda+Sequest HT. There should not be Chimerys+Amanda.
Our reply: We fixed the lines 82-83.
Lines 100-102 and 106-107: These two sentences contradict each other. After all, what was gProfiler used for, to identify proteins or to classify them.
Our reply: We fixed the lines. gProfiler is used to classify proteins.
Figure 4: Too small font makes this figure poorly readable.
Our reply: We fixed the figure 4
Lines 116-117: Why were the results for the molecular function and cellular component not also presented in table form, analogous to what was done for the biological function in Table 1.
Our reply: The EV proteins have been divided according to several biological functions not in biological processes. We fixed this error on table 1.
Table 2 and 3: What is the justification for specifying Fold Change over P Value with such a large number of decimal places?
Our reply: This is given by the analyses made with RStudio. We fixed the decimals number.
Line 189: Once again, adding up the numbers in Figure 8 comes out with a different number than 11575.
Our reply: In this case the number 11575 corresponds to the total accession number identified in the study. In the case of Figure 8, the names of genes were used to compare DIA+DDA with Exocarta and Vesciclepedia databases by omitting information. Unfortunately, the two databases are based on the name of the gene, not the accession number.
Lines 175-229: The Discussion section looks like a script for writing a proper Discussion. Brief information about 7 proteins, actually not sure why selected. You have to look only in Tables 2 and 3 that these are the proteins that had the biggest fold change. And any link between these proteins and endometrial cancer is missing.
Our reply: We added in discussion the link between these proteins and endometrial cancer.
Lines 231-242: No information is available on which of the cell lines used in the study represents type 1 and which represents type 2 endometrial cancer and metastatic or non-metastatic.
Our reply: We added information on the cell lines.
Line 239: CO2 instead of CO2 (2 in subscript).
Our reply: We fixed it.
Lines 243-249: According to ISEV recommendations, characterization of isolated EV samples should also include transmission electron microscopy (TEM) and nanoparticle tracking analysis (NTA).
Our reply: According to ISEV recommendations, we have conducted characterization of isolated EV by nanoparticle tracking analysis (NTA). We measured the concentration and diameter of the EVs. Although the TEM measures would have improved the analysis unfortunately we do not have access to the instrument.
Lines 271: H2O instead of H2O (2 in subscript).
Our reply: We fixed it.
Lines 305: EVs instead of EV.
Our reply: We fixed it.
Lines 307-308: Unnecessary repetitions: incubated overnight at 4 °C with incubated overnight at 4 °C with 1:1000 diluted primary.
Our reply: We fixed it.
Lines 309-310: No information about the companies from which the used antibodies were purchased. No information on the dilution of anti-MMP2 antibodies is available.
Our reply: We added the company from which the used antibodies were purchased. The anti-MMP2 was used 1:1000.
I have not yet encountered such a way of normalization. Rather, reference proteins, such as actin, are used in such a case.
Our reply: To normalize the results of the WB analysis, we decided to determine the total protein content of each sample by Red Ponceau, since increasing data in the literature reported that total protein normalization outperform housekeeping protein immunodetection as loading controls (B. Rivero-Gutiérrez, A. Anzola, O. Martínez-Augustin, F. Sánchez de Medina. Stain-free detection as loading control alternative to Ponceau and housekeeping protein immunodetection in Western blotting, Analytical Biochemistry, Volume 467, 2014,) (Sander H, Wallace S, Plouse R, Tiwari S, Gomes AV. Ponceau S waste: Ponceau S staining for total protein normalization. Anal Biochem. 2019 Jun 15;575:44-53.). Moreover, in EC most of the proteins commonly used as housekeeping are indeed affected and thus not adequate to be used as a control for normalization. For example, according to our previous published works, GAPDH is up-regulated in EC (Ura B, Monasta L, Arrigoni G, Franchin C, Radillo O, Peterlunger I, Ricci G, Scrimin F: A proteomic approach for the identification of biomarkers in endometrial cancer uterine aspirate. Oncotarget. 2017 Dec 12; 8: 109536–109545), while ACTB is down-regulated in the serum of EC patients (Ura B, Biffi S, Monasta L, Arrigoni G, Battisti I, Di Lorenzo G, Federico Romano F, Aloisio M, Celsi F, Addobbati R, Valle F, RampazzoE , Brucale M, ARidolfi A, Licastro D, Ricci G: Two Dimensional-Difference in Gel Electrophoresis (2D-DIGE) Proteomic Approach for the Identification of Biomarkers in Endometrial Cancer Serum. Cancers 2021, 13, 3639). Similarly, tubulin has also been found up-regulated in EC (Charles Dumontet; Mary Ann Jordan; Francis F.Y. Lee. Ixabepilone: targeting βIII-tubulin expression in taxane-resistant malignancies. Mol Cancer Ther (2009) 8 (1): 17–25). We thus decided to apply a total protein content normalization method, because we could not establish which proteins should be considered as good housekeeping in our samples.
Reviewer 1
Lines 5-7: The numbers next to the names indicating the corresponding affiliation should be placed in the superscript.
Our reply: We fixed the corresponding affiliation.
Line 6: Please add * next to the appropriate name (Monasta).
Our reply: We fixed the name Monasta.
Line 23: Please use another symbol to mark the equal contributions of two authors (V. Capaci and F. Kharrat) and add this mark next to these names in line 5.
Our reply: We fixed the names.
Lines 25-26 and line 49: In the subject literature, I have not encountered the term that EVs are cellular fragments. Please use a different term. As far as their biogenesis is concerned, cell membrane blubbing is only one of three possible ways for EVs to form. Besides, EV cargo contains a much richer set of molecules than just proteins involved in immune response and apoptosis, as the description in the Abstract suggests.
Our reply: We removed the term cellular fragments. This term was replaced and lines 25 and line 49 were fixed.
Lines 29: From the description in the methodology, it is known that EVs were isolated from the conditioned medium obtained from the culture of 4 human endometrial cancer cell lines. Thus, the term "four EV cell lines" is incorrect.
Our reply: We fixed the line 29.
Lines 35: The abbreviation EC, which was not previously defined, was introduced here. Please define it. My guess is that it refers to endometrial cancer. But it should be clarified.
Our reply: The abbreviation EC correspond to endometrial cancer. We fixed it.
Lines 45-48: No brief characterization of type 2 endometrial cancer analogous to type 1.
Our reply: We added characterization of type 2 endometrial cancer in the introduction.
Line 76: CD63 and CD9 are not markers of EVs, but of one of their subpopulations, that is, exosomes.
Our reply: We added new paragraphs describing the characterization of EVs.
.
Figure 2: According to lines 82-83 in Figure 2, there should be groups: Chimerys, 82 Chimerys+Sequest HT and Chimerys+Amanda+Sequest HT. There should not be Chimerys+Amanda.
Our reply: We fixed the lines 82-83.
Lines 100-102 and 106-107: These two sentences contradict each other. After all, what was gProfiler used for, to identify proteins or to classify them.
Our reply: We fixed the lines. gProfiler is used to classify proteins.
Figure 4: Too small font makes this figure poorly readable.
Our reply: We fixed the figure 4
Lines 116-117: Why were the results for the molecular function and cellular component not also presented in table form, analogous to what was done for the biological function in Table 1.
Our reply: The EV proteins have been divided according to several biological functions not in biological processes. We fixed this error on table 1.
Table 2 and 3: What is the justification for specifying Fold Change over P Value with such a large number of decimal places?
Our reply: This is given by the analyses made with RStudio. We fixed the decimals number.
Line 189: Once again, adding up the numbers in Figure 8 comes out with a different number than 11575.
Our reply: In this case the number 11575 corresponds to the total accession number identified in the study. In the case of Figure 8, the names of genes were used to compare DIA+DDA with Exocarta and Vesciclepedia databases by omitting information. Unfortunately, the two databases are based on the name of the gene, not the accession number.
Lines 175-229: The Discussion section looks like a script for writing a proper Discussion. Brief information about 7 proteins, actually not sure why selected. You have to look only in Tables 2 and 3 that these are the proteins that had the biggest fold change. And any link between these proteins and endometrial cancer is missing.
Our reply: We added in discussion the link between these proteins and endometrial cancer.
Lines 231-242: No information is available on which of the cell lines used in the study represents type 1 and which represents type 2 endometrial cancer and metastatic or non-metastatic.
Our reply: We added information on the cell lines.
Line 239: CO2 instead of CO2 (2 in subscript).
Our reply: We fixed it.
Lines 243-249: According to ISEV recommendations, characterization of isolated EV samples should also include transmission electron microscopy (TEM) and nanoparticle tracking analysis (NTA).
Our reply: According to ISEV recommendations, we have conducted characterization of isolated EV by nanoparticle tracking analysis (NTA). We measured the concentration and diameter of the EVs. Although the TEM measures would have improved the analysis unfortunately we do not have access to the instrument.
Lines 271: H2O instead of H2O (2 in subscript).
Our reply: We fixed it.
Lines 305: EVs instead of EV.
Our reply: We fixed it.
Lines 307-308: Unnecessary repetitions: incubated overnight at 4 °C with incubated overnight at 4 °C with 1:1000 diluted primary.
Our reply: We fixed it.
Lines 309-310: No information about the companies from which the used antibodies were purchased. No information on the dilution of anti-MMP2 antibodies is available.
Our reply: We added the company from which the used antibodies were purchased. The anti-MMP2 was used 1:1000.
I have not yet encountered such a way of normalization. Rather, reference proteins, such as actin, are used in such a case.
Our reply: To normalize the results of the WB analysis, we decided to determine the total protein content of each sample by Red Ponceau, since increasing data in the literature reported that total protein normalization outperform housekeeping protein immunodetection as loading controls (B. Rivero-Gutiérrez, A. Anzola, O. Martínez-Augustin, F. Sánchez de Medina. Stain-free detection as loading control alternative to Ponceau and housekeeping protein immunodetection in Western blotting, Analytical Biochemistry, Volume 467, 2014,) (Sander H, Wallace S, Plouse R, Tiwari S, Gomes AV. Ponceau S waste: Ponceau S staining for total protein normalization. Anal Biochem. 2019 Jun 15;575:44-53.). Moreover, in EC most of the proteins commonly used as housekeeping are indeed affected and thus not adequate to be used as a control for normalization. For example, according to our previous published works, GAPDH is up-regulated in EC (Ura B, Monasta L, Arrigoni G, Franchin C, Radillo O, Peterlunger I, Ricci G, Scrimin F: A proteomic approach for the identification of biomarkers in endometrial cancer uterine aspirate. Oncotarget. 2017 Dec 12; 8: 109536–109545), while ACTB is down-regulated in the serum of EC patients (Ura B, Biffi S, Monasta L, Arrigoni G, Battisti I, Di Lorenzo G, Federico Romano F, Aloisio M, Celsi F, Addobbati R, Valle F, RampazzoE , Brucale M, ARidolfi A, Licastro D, Ricci G: Two Dimensional-Difference in Gel Electrophoresis (2D-DIGE) Proteomic Approach for the Identification of Biomarkers in Endometrial Cancer Serum. Cancers 2021, 13, 3639). Similarly, tubulin has also been found up-regulated in EC (Charles Dumontet; Mary Ann Jordan; Francis F.Y. Lee. Ixabepilone: targeting βIII-tubulin expression in taxane-resistant malignancies. Mol Cancer Ther (2009) 8 (1): 17–25). We thus decided to apply a total protein content normalization method, because we could not establish which proteins should be considered as good housekeeping in our samples.
Reviewer 2 Report
Comments and Suggestions for Authors
The manuscript by Capaci et al has sought to identify dysregulated ECM proteins involved in endometrial cancer. Although this work is important the grouping of type 1 and type 2 cells lines only has 1 one cell line per group which is too small to deduce significant conclusions. The message of the paper is unclear and results are not justified.
Introduction/Abstract Comments
Line 29: Authors state “four EV cell lines”. This should be clarified to EV’s from 4 cell lines
Line 41: Correct “with estimated” to “with a estimated”
Line 53: Authors should expand the discussion regarding the role of EV’s in endometrial cancer
Result Comments
-Figure 1: Authors need to highlight which band is specific for CD9.
-Combine Figures 2 and 3 into one figure, also Venn diagrams should be proportioned. Title legend for “Amanda” has been cut-off.
- Figure 4: Should be separated into part A and B. Font size for table is too small and can’t read any of the values highlighted in yellow.
- Line 118: Separation of cell lines into type 1 and 2 should be addressed earlier in the text.
-Table 2/3: Values for fold change and p-value should only list two decimal points.
-Line 133: Discussion of the reactome is too limited and needs to be addressed.
- Combine figures 6 and 7.
- Figure 6: No discussion as to why these 13 cell lines are included, considering EV analysis was only conducted on 4 cell lines.
-Figure 8: Venn Diagram heading cut off
- Figure 9: Are three replicates per cell line shown (1,2,3). Ponceau is not an acceptable loading control for western blot. Please show actin or b-tubulin. Same for figure 10.
- Unsure why MMP2 was selected for further analysis than TIMP2 (top ranked protein, Table 2)
- Line 158: Where is the data from the 13 healthy controls?
- Figure 10, WB should be re-run showing all EC’s on one side and then C’s on the other.
Methods
-Were the cell lines tested for mycoplasma?
- Please list companies for antibodies
Data availability
- Unsure why the data is unavailable due to ethical reasons.
Comments on the Quality of English Language
Minor editing of text required. Some headings cut off for figures
Author Response
Reviewer 2
Line 29: Authors state “four EV cell lines”. This should be clarified to EV’s from 4 cell lines.
Our reply: Authors have clarified.
Line 41: Correct “with estimated” to “with a estimated”
Our reply: We corrected it.
Line 53: Authors should expand the discussion regarding the role of EV’s in endometrial cancer.
Our reply: The authors addressed this issue in the discussion.
Figure 1: Authors need to highlight which band is specific for CD9.
Our reply: The two band correspond to CD9.
Combine Figures 2 and 3 into one figure, also Venn diagrams should be proportioned. Title legend for “Amanda” has been cut-off.
Our reply: Is not possible to combine the figures 2 and 3 because they contain different information. We fixed the title legend for “Amanda”.
Figure 4: Should be separated into part A and B. Font size for table is too small and can’t read any of the values highlighted in yellow.
Our reply: The figures were separated according to the reviewer indications.
Table 2/3: Values for fold change and p-value should only list two decimal points.
Our reply: We corrected the fold change and p-value values.
Line 133: Discussion of the reactome is too limited and needs to be addressed.
Our reply: We expand the discussion on the Reactome.
Combine figures 6 and 7.
Our reply: We thank the reviewer for the suggestion, but the figures contain different information.
Figure 6: No discussion as to why these 13 cell lines are included, considering EV analysis was only conducted on 4 cell lines.
Our reply: Firstly, we want to clarify that the 13 EC samples are not cell lines but EC tissue samples. We used these tissues to validate that MMP2 levels in EVs were modulated not only in cell lines but even in primary tissue from patients. We expand the discussion how the reviewer suggest.
Figure 8: Venn Diagram heading cut off
Our reply: We fixed the figure.
Figure 9: Are three replicates per cell line shown (1,2,3). Ponceau is not an acceptable loading control for western blot. Please show actin or b-tubulin. Same for figure 10.
Our reply: As we responded to the first reviewer, we chose the Red Ponceau for normalization because since increasing data in the literature reported that total protein normalization outperform housekeeping protein immunodetection as loading controls (B. Rivero-Gutiérrez, A. Anzola, O. Martínez-Augustin, F. Sánchez de Medina. Stain-free detection as loading control alternative to Ponceau and housekeeping protein immunodetection in Western blotting, Analytical Biochemistry, Volume 467, 2014,) (Sander H, Wallace S, Plouse R, Tiwari S, Gomes AV. Ponceau S waste: Ponceau S staining for total protein normalization. Anal Biochem. 2019 Jun 15;575:44-53.). Moreover, in EC most of the proteins commonly used as housekeeping are indeed affected and thus not adequate to be used as a control for normalization. For example, according to our previous published works, GAPDH is up-regulated in EC (Ura B, Monasta L, Arrigoni G, Franchin C, Radillo O, Peterlunger I, Ricci G, Scrimin F: A proteomic approach for the identification of biomarkers in endometrial cancer uterine aspirate. Oncotarget. 2017 Dec 12; 8: 109536–109545), while ACTB is down-regulated in the serum of EC patients (Ura B, Biffi S, Monasta L, Arrigoni G, Battisti I, Di Lorenzo G, Federico Romano F, Aloisio M, Celsi F, Addobbati R, Valle F, RampazzoE , Brucale M, ARidolfi A, Licastro D, Ricci G: Two Dimensional-Difference in Gel Electrophoresis (2D-DIGE) Proteomic Approach for the Identification of Biomarkers in Endometrial Cancer Serum. Cancers 2021, 13, 3639). Similarly, tubulin has also been found up-regulated in EC (Charles Dumontet; Mary Ann Jordan; Francis F.Y. Lee. Ixabepilone: targeting βIII-tubulin expression in taxane-resistant malignancies. Mol Cancer Ther (2009) 8 (1): 17–25). We thus decided to apply a total protein content normalization method, because we could not establish which proteins should be considered as good housekeeping in our samples.
Unsure why MMP2 was selected for further analysis than TIMP2 (top ranked protein, Table 2)
Our reply: In addition to being one of the two most abundant proteins MMP2 is also an enzyme that plays a key role in ECM degradation in tumor cell growth, regulation of tumor angiogenesis, immune surveillance, invasion and metastasis.
Line 158: Where is the data from the 13 healthy controls?
Our reply: We added the data for the 13 healthy controls.
Figure 10, WB should be re-run showing all EC’s on one side and then C’s on the other.
Our reply: We want to thank the reviewer for the suggestion. We decided to proceed due to the number of samples, with the placement of healthy samples with tumor samples because it is easier to understand the differences in this way. This is accepted from journals (Qiuli Yu , Liqin Xu , Long Chen, Baier Sun , Zhiyun Yang, Kunqin Lu, Zhiyong Yang . Vinculin expression in non-small cell lung cancer. J Int Med Res
. 2020 Jan;48(1):300060519839523), (Bo Wang , Chaoyang Liang , Huifeng Liu , Jixing Lin , Bailin Wang , Kaijie Fan , Zhipeng Ren, Bin Wang, Tong Li, Kang Qi, Xiaodong Tian. The expression of mouse double minute 2 homolog and P73 had no correlation with growth arrest DNA damage-inducible gene 45α in patients with non-small-cell lung carcinoma: A STROBE-compliant study. Medicine (Baltimore). 2019 Dec;98(51):e17944), (Valeria Capaci, Lorenzo Monasta, Michelangelo Aloisio, Eduardo Sommella, Emanuela Salviati, Pietro Campiglia, Manuela Giovanna Basilicata, Feras Kharrat , Danilo Licastro , Giovanni Di Lorenzo, Federico Romano, Giuseppe Ricci, Blendi Ura. A Multi-Omics Approach Revealed Common Dysregulated Pathways in Type One and Type Two Endometrial Cancers. Int J Mol Sci. 2023 Nov 7;24(22):16057). Unfortunately, at this time we don’t have enough sample to repeat the experiments.
Were the cell lines tested for mycoplasma?
Our reply: The cell lines were all tested for mycoplasma by PCR. The test results were all negative.
Please list companies for antibodies.
Our reply: We added companies list for antibodies.
Unsure why the data is unavailable due to ethical reasons.
Our reply: The data are not publicly available due to ethical reasons.
Reviewer 3 Report
Comments and Suggestions for Authors
The article titled "The deep proteomics approach identified extracellular vesicular proteins correlated to extracellular matrix in type one and two endometrial cancer," presented by Capaci et al., conducts an analysis aimed at identifying dysregulated ECM proteins in EV and finding possible correlations to key pathways for tumor development in endometrial cancer through the analysis of published proteomic data.
While the article is intriguing and the authors attempt to experimentally validate the results of deep proteomics using samples from 13 patients with type 1 and type 2 EC and 13 endometrial samples by determining the abundance of MMP2 and its counterpart TIMP2, there are some observations that could complement the article.
Observations-
-
I suggest that validating the detection of MMP2 through zymography assays with substrate would be beneficial, as it could help determine the enzymatic function of MMP2 in exosomes in cell cultures. Additionally, using specific inhibitors like MMP-2 Inhibitor I, cis-9-Octadecenoyl-N-hydroxylamide, could be considered. It has been reported that the tumor microenvironment can modulate the activity of these MMPs, and establishing a correlation between protein abundance and potential activity would be insightful.
-
Complete blot images are necessary to visualize the size of MMP2 and other detected proteins.
-
Is there a possibility of using a constitutive or housekeeping protein to normalize control data compared to EC? While the difficulty of using some metabolic proteins due to their differential expression is known, utilizing one of these proteins would help validate the results. see
Khan, S., Varricchio, A., Ricciardelli, C., & Yool, A. J. (2023). Invasiveness of endometrial cancer cell lines is potentiated by estradiol and blocked by a traditional medicine Guizhi Fuling at clinically relevant doses. Frontiers in oncology, 12, 1015708. https://doi.org/10.3389/fonc.2022.1015708
-
The discussion should focus more on the potential functions of MMP2, TIMP 2, and avoid redundancy with the description of results.
-
With the suggested results from point 1, creating an illustrative figure depicting the function of MMP2 in EC would be beneficial.
Author Response
Reviewer 3
I suggest that validating the detection of MMP2 through zymography assays with substrate would be beneficial, as it could help determine the enzymatic function of MMP2 in exosomes in cell cultures. Additionally, using specific inhibitors like MMP-2 Inhibitor I, cis-9-Octadecenoyl-N-hydroxylamide, could be considered. It has been reported that the tumor microenvironment can modulate the activity of these MMPs, and establishing a correlation between protein abundance and potential activity would be insightful.
Our reply: We thank the reviewer for this observation. We want to clarify that aim of our study isn’t on the enzymatic function of MMP2 in exosomes. This is a proteomics study to described protein component of EV and to identify and quantify the ECM proteins in endometrial cancer EV. To take into account the suggestion of the reviewer, MMP2 mechanism will be the next study on EC.
Complete blot images are necessary to visualize the size of MMP2 and other detected proteins.
Our reply: We added like supplement file with the complete blot images.
Is there a possibility of using a constitutive or housekeeping protein to normalize control data compared to EC? While the difficulty of using some metabolic proteins due to their differential expression is known, utilizing one of these proteins would help validate the results.
Our reply: We have already addressed this topic with reviewer 1 and 2. However we reply to the reviewer. We chose the Red Ponceau for normalization because since increasing data in the literature reported that total protein normalization outperform housekeeping protein immunodetection as loading controls (B. Rivero-Gutiérrez, A. Anzola, O. Martínez-Augustin, F. Sánchez de Medina. Stain-free detection as loading control alternative to Ponceau and housekeeping protein immunodetection in Western blotting, Analytical Biochemistry, Volume 467, 2014,) (Sander H, Wallace S, Plouse R, Tiwari S, Gomes AV. Ponceau S waste: Ponceau S staining for total protein normalization. Anal Biochem. 2019 Jun 15;575:44-53.). Moreover, in EC most of the proteins commonly used as housekeeping are indeed affected and thus not adequate to be used as a control for normalization. For example, according to our previous published works, GAPDH is up-regulated in EC (Ura B, Monasta L, Arrigoni G, Franchin C, Radillo O, Peterlunger I, Ricci G, Scrimin F: A proteomic approach for the identification of biomarkers in endometrial cancer uterine aspirate. Oncotarget. 2017 Dec 12; 8: 109536–109545), while ACTB is down-regulated in the serum of EC patients (Ura B, Biffi S, Monasta L, Arrigoni G, Battisti I, Di Lorenzo G, Federico Romano F, Aloisio M, Celsi F, Addobbati R, Valle F, RampazzoE , Brucale M, ARidolfi A, Licastro D, Ricci G: Two Dimensional-Difference in Gel Electrophoresis (2D-DIGE) Proteomic Approach for the Identification of Biomarkers in Endometrial Cancer Serum. Cancers 2021, 13, 3639). Similarly, tubulin has also been found up-regulated in EC (Charles Dumontet; Mary Ann Jordan; Francis F.Y. Lee. Ixabepilone: targeting βIII-tubulin expression in taxane-resistant malignancies. Mol Cancer Ther (2009) 8 (1): 17–25). We thus decided to apply a total protein content normalization method, because we could not establish which proteins should be considered as good housekeeping in our samples.
The discussion should focus more on the potential functions of MMP2, TIMP 2, and avoid redundancy with the description of results.
Our reply: We expand the discussion as the reviewer suggests.
With the suggested results from point 1, creating an illustrative figure depicting the function of MMP2 in EC would be beneficial.
Our reply: The same answer as the first point.
Reviewer 4 Report
Comments and Suggestions for Authors
This study focuses on identifying dysregulated extracellular matrix (ECM) proteins in extracellular vesicles (EVs) from endometrial cancer cells using a proteomic approach. Despite offering valuable insights into ECM protein dysregulation in EC, the study presents significant shortcomings in its description and methodological approaches, limiting the comprehensiveness of its findings.
The introduction does not sufficiently justify the goals and objectives of the research. In particular, there is almost no description of the role of the extracellular matrix in the development of the oncologic process, and the association of this process with extracellular vesicles. Can the proteins of the extracellular matrix be deliberately loaded into extracellular vesicles? Similarly, the role of protein cargo in extracellular vesicles in the pathogenesis of oncological diseases has not been considered. Also, there is no justification for the use of cell lines for comparison with primary cultures of patients.
Results
According to the MISEV 2023 requirements, markers used for characterizing extracellular vesicles should include vesicle markers, as well as cytoplasmic proteins that are not loaded into vesicles, and a quantitative analysis of extracellular vesicles with the determination of their size is also required. The presented analysis is not sufficient to characterize the isolated preparation as "extracellular vesicles."
Figure 2 requires detailed description.
It is necessary to provide a morphological characterisation of the obtained cell cultures from patients and to perform their phenotyping.
In Figure 4, the image on the right is unreadable due to low magnification.
Method
It is necessary to describe the principle of the method for isolating extracellular vesicles, justify its choice, and explain what impurities, apart from vesicles, may be isolated using this approach. It is well known that the method used by the authors for isolating vesicles leads to significant contamination with lipoproteins and protein impurities. The choice of this method requires explanation and should be discussed as a limitation of this study (Brennan, K., Martin, K., FitzGerald, S.P., et al. A comparison of methods for the isolation and separation of extracellular vesicles from protein and lipid particles in human serum. Sci Rep 10, 1039 (2020). https://doi.org/10.1038/s41598-020-57497-7; Patel, G.K., Khan, M.A., Zubair, H., et al. Comparative analysis of exosome isolation methods using culture supernatant for optimum yield, purity and downstream applications. Sci Rep 9, 5335 (2019). https://doi.org/10.1038/s41598-019-41800-2).
It is necessary to present a table with the clinico-anamnestic characteristics of the patients, including the exact diagnosis according to ICD-11.
In the methods section, there is a subsection “4.3. Patients”, which describes the patients, however, further description of how the primary cell cultures were obtained or other manipulations with this cohort of patients is missing.
Discussion
The discussion is superficial and does not address the main research questions.
It is necessary to discuss and justify the choice of studied cell cultures, justify their necessity when primary patient material was used.
A comparison of MMP2 levels with subsequent catamnestic observations of patients should be conducted, including the assessment of the metastasis process.
Author Response
Reviewer 4.
The introduction does not sufficiently justify the goals and objectives of the research. In particular, there is almost no description of the role of the extracellular matrix in the development of the oncologic process, and the association of this process with extracellular vesicles. Can the proteins of the extracellular matrix be deliberately loaded into extracellular vesicles? Similarly, the role of protein cargo in extracellular vesicles in the pathogenesis of oncological diseases has not been considered. Also, there is no justification for the use of cell lines for comparison with primary cultures of patients.
Our reply: We thank the reviewer for this good suggestion, we modified the introduction discussing the role ECM, Tumor microenvironment and EVs. Indeed, to investigate the role of EVs released by endometrial cancer cells we exploited EC cell lines models, both due to difficulty to obtain fresh samples from patients (in particular for type II ECs), and to obtain EVs whose origin was unequivocally from the tumor cells and not from the tumor microenvironment or other tissues. these considerations were now added in discussion section.
According to the MISEV 2023 requirements, markers used for characterizing extracellular vesicles should include vesicle markers, as well as cytoplasmic proteins that are not loaded into vesicles, and a quantitative analysis of extracellular vesicles with the determination of their size is also required. The presented analysis is not sufficient to characterize the isolated preparation as "extracellular vesicles."
Our reply: In addition to CD9 and CD63 we added other marker (HSP90 α/β, HSC70) used for characterizing extracellular vesicles accepted from MISEV 2023. As the reviwer suggest for identification of cytoplasmic proteins not loaded into vesicles we selected 5576 cytoplasmic proteins from a Uniprot proteomics database with 19304 proteins. We compared these proteins with the proteins identified in DIA and DDA. We identified 2639 cytoplasmic proteins not loaded into vescicles (See Supplemental data). We carried out analysis with the NTA to determinate the concentration and the dimensions of the EVs. For this purpose we added 2 new paragraphs in the manuscript.
Figure 2 requires detailed description.
Our reply: We described more in detail Figure 2.
It is necessary to provide a morphological characterization of the obtained cell cultures from patients and to perform their phenotyping.
Our reply: for the present study we didn’t use primary patient cell lines, just tissue. The only cell lines we’ve used in the study (ISHIKAWA, HEC1A, KLE, ANCA3) that are commercial cell lines.
In Figure 4, the image on the right is unreadable due to low magnification.
Our reply: We fixed the figure 4.
It is necessary to present a table with the clinico-anamnestic characteristics of the patients, including the exact diagnosis according to ICD-11.
Our reply: We added Supplementary Table S1, where we describe the clinical and pathological characteristics of the patient and the ICD-11.
In the methods section, there is a subsection “4.3. Patients”, which describes the patients, however, further description of how the primary cell cultures were obtained or other manipulations with this cohort of patients is missing.
Our reply: In this study we didn’t use primary cell cultures.
A comparison of MMP2 levels with subsequent catamnestic observations of patients should be conducted, including the assessment of the metastasis process.
Our reply: Following the Reviewer’s suggestion, we compared the levels of MMP2 with age, histotype, grade, tumor size, and stage. However, numbers are too small to be able to result in statistical significance, even for those analyses that showed some weak associations or rank correlations. We added a note to the manuscript stating that it would be interesting to increase the numbers in order to investigate associations between MMP2 and descriptive and subsequent catamnestic observations. No patients have developed metastases so far.
It is necessary to describe the principle of the method for isolating extracellular vesicles, justify its choice, and explain what impurities, apart from vesicles, may be isolated using this approach. It is well known that the method used by the authors for isolating vesicles leads to significant contamination with lipoproteins and protein impurities. The choice of this method requires explanation and should be discussed as a limitation of this study (Brennan, K., Martin, K., FitzGerald, S.P., et al. A comparison of methods for the isolation and separation of extracellular vesicles from protein and lipid particles in human serum. Sci Rep 10, 1039 (2020). https://doi.org/10.1038/s41598-020-57497-7; Patel, G.K., Khan, M.A., Zubair, H., et al. Comparative analysis of exosome isolation methods using culture supernatant for optimum yield, purity and downstream applications. Sci Rep 9, 5335 (2019). https://doi.org/10.1038/s41598-019-41800-2).
Our reply: A large amount method, based on different principles have been developed for EVs and exosome isolation, each of these carrying advantages and disadvantages, as previously reported by Patel et al. In this work we adopted a commercial kit optimized for Exosome isolation. Measurement of vesicle diameter by NTA revealed as population of EVs, in the size range of exosomes (116.4 nM in average). Indeed, we also performed immunoblot of exosomal marker CD9 and CD63. Then based on marker presence and vesicles dimension we can surely affirm the isolated vesicles are in bona fide exosome. These aspects were now added in discussion.
The discussion is superficial and does not address the main research questions.
It is necessary to discuss and justify the choice of studied cell cultures, justify their necessity when primary patient material was used.
Our reply: We agree with the reviewer, indeed the discussion have been largely modified adding the rational, justifying the choose of cell models and explaining the main goals of the study.
Round 2
Reviewer 1 Report
Comments and Suggestions for Authors
The authors responded to all my comments mentioned in my review and brought to the manuscript the necessary changes that would definitely enhance its quality.
Evaluating a manuscript that is in 'marked' form is quite difficult. It seems to me that it is still necessary to review it for linguistic correctness and typos.
Comments on the Quality of English LanguageEvaluating a manuscript that is in 'marked' form is quite difficult. It seems to me that it is still necessary to review it for linguistic correctness and typos.
Author Response
Evaluating a manuscript that is in 'marked' form is quite difficult. It seems to me that it is still necessary to review it for linguistic correctness and typos.
Our reply: We fixed the language.
Reviewer 2 Report
Comments and Suggestions for Authors
Not all reviewer concerns have been addressed appropriately.
1. Details of mycoplasma testing must be included in the methods section
2. Data Availability Statement: Authors have not provided sufficient details as to why data cannot be publicly available due to ethical reasons. Which data are you referring to?
Figure 4: As stated previously, highlighting in yellow makes the text unreadable. Choose a different highlight color or use black font.
Comments on the Quality of English LanguageFine. Minor corrections required
Author Response
Details of mycoplasma testing must be included in the methods section.
Our reply: We added the details of mycoplasma testing.
Data Availability Statement: Authors have not provided sufficient details as to why data cannot be publicly available due to ethical reasons. Which data are you referring to?
Our reply: We would like to clarify that all supplementary data will be available when the manuscript is published.
Figure 4: As stated previously, highlighting in yellow makes the text unreadable. Choose a different highlight color or use black font.
Our reply: The Table from Figure 4 is generated automatically by the bioinformatic tool gProfiler. Is not possible to change this table.
Reviewer 3 Report
Comments and Suggestions for Authors
The authors have made significant changes to the data analysis and discussion which improve the article. I consider their responses to be adequate as they mention that some points will be addressed in future research with MMP7.
As an observation, the authors again do not submit the original blots with the molecular weight markers to corroborate the MW of MMP2 in the extracts of cell lines and patient tissue, nor is it mentioned what each Ponceau-stained gel corresponds to. This would be my only observation.
Author Response
As an observation, the authors again do not submit the original blots with the molecular weight markers to corroborate the MW of MMP2 in the extracts of cell lines and patient tissue, nor is it mentioned what each Ponceau-stained gel corresponds to. This would be my only observation.
Our reply: We submit original blots with the molecular weight markers of MMP2 and each Ponceau-stained gel corresponds to the cells and tissue proteins.
Reviewer 4 Report
Comments and Suggestions for Authors
I am pleased to inform you that the authors have responded to all my comments and have submitted a revised version of the manuscript, which now has improved scholarly wording. I have no further comments and believe that the manuscript is ready for publication in its current form.
Author Response
We thank the reviewer for his suggestions.